# Climate change overtakes coastal engineering as the dominant driver of hydrological change in a large shallow lagoon

Peisheng Huang[1,2], Karl Hennig[3], Jatin Kala[4], Julia Andrys[4], Matthew R. Hipsey[1,2]

[1]Aquatic Ecodynamics, UWA School of Agriculture and Environment, The University of Western Australia, Crawley WA 6009, Australia.
[2]UWA Oceans Institute, The University of Western Australia, Crawley WA 6009, Australia.
[3]Water Science Branch, Department of Water and Environmental Regulation, Perth WA 6842, Australia.
[4]Environmental and Conservation Sciences, Murdoch University, Murdoch WA 6015, Australia.

*Correspondence to*: Peisheng Huang (peisheng.huang@uwa.edu.au)

**Abstract.** Ecosystems in shallow, micro-tidal lagoons are particularly sensitive to hydrologic changes. Lagoons are complex transitional ecosystems between land and sea, and the signals of direct human disturbance to the lagoon can be confounded by variability of the climate system, but from an effective estuary management perspective the effects of climate versus direct human engineering interventions need to be identified separately. This study developed a 3D finite-volume hydrodynamic model to assess changes in hydrodynamics of the Peel-Harvey Estuary, a large shallow lagoon with restricted connection with ocean, considering how attributes such as water retention time, salinity and stratification have responded to a range of factors, focusing on the drying climate trend and the opening of a large artificial channel over the period from 1970 to 2016, and how they will evolve under current climate projections. The results show that the introduction of the artificial channel has fundamentally modified the flushing and mixing within the lagoon, and the drying climate has changed the hydrology by comparable magnitudes to that of the opening of the artificial channel. The results also highlight the complexity of their interacting impacts. Firstly, the artificial channel successfully improved the estuary flushing by reducing average water ages by 20-110 days; while in contrast the reduced precipitation and catchment inflow had a gradual opposite effect on the water ages, and during the wet season this has almost counteracted the reduction brought about by the channel. Secondly, the drying climate caused an increase in the salinity of the lagoon by 10-30 PSU; whilst the artificial channel increased the salinity during the wet season, it has reduced the likelihood of hypersalinity (>40 PSU) during the dry season in some areas. The opening of the artificial channel was also shown to increase the seawater fluxes and salinity stratification, while the drying climate acted to reduce the salinity stratification in the main body of the estuary. The impacts also varied spatially in this large lagoon. The southern estuary, which has the least connection with ocean through the natural channel, is the most sensitive to climate change and the opening of the artificial channel. The projected future drying climate is shown to slightly increase the retention time and salinity in the lagoon, and increase the hypersalinity risk in the rivers. The significance of these changes for nutrient retention and estuary ecology are discussed, highlighting the importance of these factors when setting up monitoring programs, environmental flow strategies and nutrient load reduction targets.

## 1 Introduction

Hydrologic features such as water circulation and retention, and the pattern of saline water intrusion, are critical in shaping estuarine ecosystems. The interactions between freshwater runoff pulses with ocean water can create complex hydrodynamics that subsequently structures coastal biogeochemical processes, including the distribution of sediment and nutrients, and areas favourable for primary productivity (e.g. Legović et al., 1994; Kasai et al., 2010; Watanabe et al., 2014; Cloern et al., 2017). Whilst nutrient loads are the primary determinant affecting the long-term trophic state of coastal waters (Howarth & Marino, 2006; Williamson et al., 2017), the time-scales associated with water retention and mixing are critical in mediating the relationship between nutrient inputs and the ensuing water quality response, including the likelihood of nuisance algal blooms

or hypoxia (e.g. Knoppers et al., 1991; Ferreira et al., 2005; Paerl et al., 2006; Zhu et al., 2017). The retention of water and hydrodynamic patterns that emerge in any given site are largely dependent upon local geomorphological features, though increasingly coastal engineering and changes in river hydrology disturb natural patterns of water exchange (Knoppers et al., 1991; Kjerfve et al., 1996; Dufour et al., 2001; Gong et al., 2008; Odebrecht et al., 2015; Almroth-Rosell et al., 2016). Understanding and predicting these hydrologic changes is critical to underpin adaptive approaches to estuary water quality management and ecological restoration.

Coastal lagoons and embayments with low rates of ocean exchange are particularly sensitive relative to other estuary forms. The typical low flushing rates leads to high rates of deposition of sediment and particulate matter, and accumulation of nutrients (e.g. Newton et al., 2014, 2018; Paerl et al., 2014). They are also productive ecosystems and often experience conflicting interests between the ecosystem services they provide and the pressures from urban development and agricultural expansion (Petersen et al., 2008; Zaldívar et al., 2008; Pérez-Ruzafa et al., 2011; Basset et al., 2013; Newton et al., 2014). In most cases, salt intrusion mediates lagoon salinity and drives a difference between the surface and bottom salinity (salinity stratification). In highly seasonal systems this effect leads to notable oxygen depletion and establishes hypoxia in the bottom boundary layer (Bruce et al., 2014; Cottingham et al., 2014; Huang et al., 2018). In Mediterranean climate regions, further concerns of hypersalinity through evaporation during the long dry summer and autumn months also exist (Potter et al., 2010). As a result, the hydrodynamics of coastal lagoons are frequently modified through the creation of artificial channels built to enhance hydrologic connectivity to the ocean, and increase nutrient export (e.g. Breardley 2005; Manda et al., 2014; Prestrelo and Monteiro-Neto, 2016), or indirectly by engineering projects associated with dredging and coastal management (e.g. Ghezzo et al., 2010; Sahu et al., 2014).

Changes in lagoon hydrology result from variability in river flows, and meteorological and ocean conditions, alongside (sporadic) human interventions associated with coastal engineering developments. Globally, many studies have shown that coastal lagoon systems are vulnerable to climate change, including the factors from reduced flow and/or sea level rise (Nicholls & Hoozemans, 1996; Nicholls et al., 1999; Scavia et al., 2002; Chapman, 2012; Newton et al., 2014; Umgiesser et al., 2014). In particular, shallow coastal lagoons respond quickly to the ocean and catchment inputs whilst their geomorphological characteristics (bathymetry and especially the configuration of their inlets with the open sea) affect their hydrodynamics including circulation patterns, flushing time and water mixing (e.g. Smith, 1994; Spaulding, 1994; Koutitonsky, 2005; Umgiesser et al., 2014). This attribute has meant that these systems therefore amplify the salinity and temperature changes expected from climate change relative to the open sea, and that they can serve as sentinel systems for global change studies (Ferrarin et al., 2014). On the other hand, anthropogenic activities introduce hydrological modifications associated with water resource management (e.g. Hollis, 1990; Kingsford et al., 2006; Gong et al., 2008) and engineering modifications (Ghezzo et al., 2010; Garcia-Oliva et al., 2018). Among which, the opening of artificial channels in the lagoon systems has been a popular measure to enhance the ocean connectivity, but this has the effect to fundamentally alter the hydrology and aquatic communities (e.g. Lord, 1998; Manda et al., 2014; Prestrelo and Monteiro-Neto, 2016; Garcia-Oliva et al., 2018). Changes in the connection of restricted lagoons with the ocean can exhibit a marked change in the salinity pattern or the extent of hypersaline conditions (Kjerfve, 1994; Gamito et al., 2005), and subsequently influence the ecosystem within these lagoons (Gamito, 2006; Garcia-Oliva et al., 2018). These enhance the need to assess lagoon hydrological function if these impacts are to be predicted and, where necessary, mitigation measures developed, in conjunction with climate influences. However, tracking the long-term changes in hydrology remains an ongoing challenge since the signals of human disturbance are often confounded by variability of the climate system and lost in the dynamic estuarine conditions (Feyrer et al., 2015; Cloern et al., 2016), and the interacting effects of introducing human interventions in conjunction with climate change trends are not

necessarily easy to predict. For example, the opening of an artificial channel and a drying climate can both introduce more ocean water into an estuary. On the other hand, the drying climate enhances water residence time, which may cancel out flushing benefits from the artificial channel. The combined effects are further complicated in large lagoon-type estuaries with complex morphology, where complicated patterns of water retention and stratification can develop (e.g. Ferrarin et al., 2013). From a lagoon management perspective, it is necessary to attribute the impacts from climate and human activity factors to better plan the necessary estuary and catchment management activities, including adaptation strategies associated with nutrient load targets and, in some cases, environmental water provision.

Here we explore these ideas through reconstruction of the long-term hydrologic evolution of a large estuarine lagoon in Western Australia: the Peel-Harvey Estuary (PHE). The PHE system has been subject to both a notable drying climate trend and substantial coastal modification in the form of an opening of a large artificial channel, coastal development and dredging. The artificial channel, termed the "Dawesville Cut" (hereafter referred as "the Cut"), was built in 1994 with the purpose of increasing the flushing and reducing nutrient concentrations. In parallel, the impact on inland water resources of recent climate trends has been particularly acute in the PHE catchment, which was acknowledged by the IPCC AR4 identifying this region as one that has experienced amongst the greatest impact on divertible water resources in the world (Izrael et al., 2007; Bates et al., 2008). From the 1970s, rainfall has decreased by 16% and stream flows have declined by more than 50%, a trend which has appeared to accelerate since the 2000s (Silberstein et al., 2012). Whilst the nutrient and phytoplankton concentrations have been successfully reduced by the construction of the channel (Brearley, 2005), the long-term river flows have shown a clear trend of decreasing inputs to the estuary with concerns for the conditions of the tidal riverine portions of the system (Gillanders et al., 2011; Hallett et al., 2018). A series of water quality improvement plans (e.g. Environmental Protection Authority, 2008; Rogers et al., 2010) continue to be developed to promote estuary health, however, ongoing concerns about the current and future water quality and ecologic condition of the system (Valesini et al., 2019) requires knowledge of spatiotemporal changes in water retention, stratification and salinisation to support adaptation efforts.

It is therefore the aim of this study to develop a methodology to disentangle drivers of change of the PHE system, over the period from 1970 to 2016, and outline the expected future trajectory of lagoon conditions. To this end we employ a 3-dimentional finite-volume hydrodynamic model for analysis of environmental drivers on estuarine hydrology by comparing current and counter-factual modelling scenarios to enable attribution of the drivers of change. To enable the long-term reconstruction of the model simulations for periods before the instrument record, and for future conditions, we drive the model with a hybrid set of weather and hydrological boundary condition data from observations and supporting models. The results of simulations are presented to analyse the sensitivity of water retention time, salinity and stratification within the lagoon to selected factors. By untangling the effect of the drying climate versus the Cut opening, through time and space, we explore the results through the lens of nutrient load reduction targets and biodiversity management implications. We anticipate the approach adopted here can be useful to assist in the climate change adaptation efforts for other estuarine lagoons in mid-latitude regions.

## 2 Methods

### 2.1 Site description

PHE is a large shallow coastal estuary-lagoon system located approximately 75 km south of Perth in Western Australia (Figure 1), which is listed under the Ramsar convention for wetlands of international significance. The estuary has a complex

morphometry and comprises two shallow lagoons, one is the Peel Inlet, a circular inlet to the north, and the other the Harvey estuary, an oblong lagoon attached to the Peel Inlet at its north-eastern edge, with a combined area of approximately 133 $km^2$. The estuary connects to the ocean via two channels: 1) the Mandurah Channel, a natural but narrow 5 km long channel with water depths varying between 2 m and 5 m; and 2) the Cut, an artificial channel of about 2.5 km long, 200 m wide and between 6 and 6.5 m deep, built in 1994 (Bicknell, 2006; Environmental Protection Authority, 2008). The system experiences a Mediterranean-type climate characterised by a strong seasonal pattern of cool wet winters and hot dry summers, with almost all of the annual rainfall occurring during the cooler months of May to October (Gentilli, 1971; Finlayson & McMahon, 1988). The estuary experiences a micro-tidal regime, with a range < 1m. The tide is dominated by the lunar diurnal constituents (K1, O1) contributing 87% of the tide potential energy, followed by the solar diurnal constituent (P1), principal lunar semidiurnal constituent (M2), and principal solar semi-diurnal constituent (S2) (Table 1). The coastal catchment of the estuary is drained by three major river systems: the Serpentine, Murray and Harvey Rivers (on average contributing 16.4%, 46.5%, and 30.8% to the total flow, respectively), and numerous minor drains (contributing 6.3% to the total flow) (Kelsey et al., 2011).

**Table 1.** Principal tidal constituents for Fremantle tide record from 1970 to 2017

| Constituents | Potential Energy (%) | Amplitude (m) | Greenwich phase lag (degrees) | Frequency (cycles per hour) |
|---|---|---|---|---|
| K1 | 61.08 | 0.156 | 324 | 0.0418 |
| O1 | 25.95 | 0.101 | 308 | 0.0387 |
| P1 | 6.06 | 0.049 | 314 | 0.0416 |
| M2 | 3.53 | 0.0374 | 323 | 0.0805 |
| S2 | 3.37 | 0.0365 | 334 | 0.0833 |

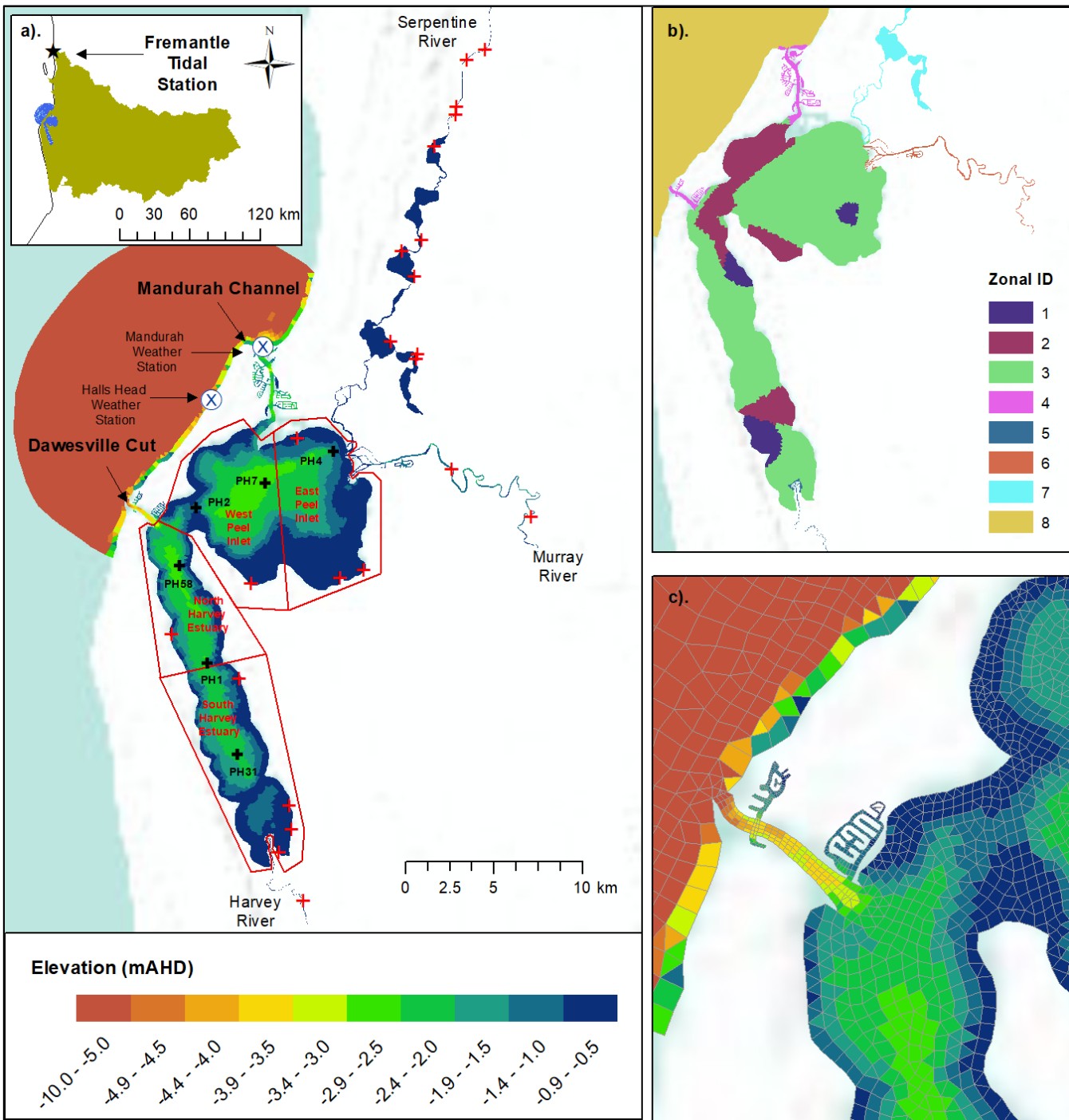

139

**Figure 1.** (a) Model domain of the Peel-Harvey Estuary and three main rivers: Serpentine River, Murray River, and Harvey River, the tidal portion of each is depicted up to the gauge location. The colours indicate the water depths of the study domain in unit of Australian Height Datum (AHD); the black crosses within the estuary indicate the locations of 6 monitoring sites, the red crosses indicate the locations of ungauged flows entering the main rivers and estuary; the blue crosses indicate the location of weather stations, and the red polygons indicate the areas for result analysis. b) Zonal categorization of the model domain according to the area and aquatic vegetation biomass (see Table 2), and c) a zoom-in view of the artificial channel Dawesville Cut, constructed in 1994 to improve ocean flushing.

147

148

**2.2 Modelling Approach**

**2.2.1 Hydrodynamic modelling platform and numerical methods**

The TUFLOW-FV (BMT WBM, 2013) hydrodynamic model was adopted, using a flexible-mesh (finite volume) approach to resolve the variations in water level, horizontal salinity distribution and vertical density stratification in response to tides, inflows and surface thermodynamics. The mesh consists of triangular and quadrilateral elements of different size that are suited to simulating areas of complex estuarine morphometry. To meet accuracy requirements, fine-grid resolution (mean mesh area ~12000 $m^2$) was used within the lagoons and coarse resolution was implemented towards the ocean boundary. The vertical mesh discretization adopted a hybrid sigma-z coordinate allowing multiple surface Lagrangian layers to respond to tidal elevation changes. The layer thickness was 0.2 m at depths of 1.0 – 5.0 m that gradually increased to 0.5 m in deeper water and then five uniformly-distributed sigma layers were added above the fixed-thickness layers. The finite volume numerical scheme solves the conservative integral form of the nonlinear shallow water equations in addition to the advection and transport of scalar constituents such as salinity and temperature. The equations are solved in 3D with baroclinic coupling from both salinity and temperature using the UNESCO equation of state (Fofonoff & Millard, 1983). The water level at the ocean boundary was specified with the record (every 15 minutes) from the Fremantle gauge station located about 52 km to the north of the study site, while the velocity was calculated internally based on a radiation condition assumption.

Surface momentum exchange and heat dynamics are solved internally within TUFLOW-FV. In the current application, turbulent mixing of momentum and scalars has been calculated using the Smagorinsky scheme in the horizontal plane and through coupling with the General Ocean Turbulence Model (GOTM) for vertical mixing. The bottom shear stress was calculated using a roughness-length relationship assuming a rough-turbulent logarithmic velocity profile in the lowest model layer. The roughness length, $z_0$, settings were based on the area type (e.g. coast, rivers, and estuary) and the estimated biomass of aquatic vegetation within the cell. For this purpose, the modelling bottom was categorized into eight zones (Figure 1b) where the benthic characteristics and associated $z_0$ in each zone were specified (Table 2). While the setting of $z_0$ affected the water advection and uncertainty remains in the spatial (and temporal) variability in the $z_0$, it is important to note that the modelled water age and salinity do not change fundamentally over a reasonable range of $z_0$, as shown in the results of the model sensitivity tests described later.

Multiple concepts of hydrodynamic time parameters (flushing time, residence time, water age, export time, etc.) have been used in coastal hydrology research, and each of these parameters are different in their definition and application (e.g. Monsen et al., 2002; Jouon et al., 2006; Sheldon & Alber, 2006). This study employed a few hydrodynamic time parameters to serve for different study purposes. The first time parameter was the 'water age', which was defined as the time the water had spent since entering the estuary through the boundaries (either the ocean or rivers), and was computed in each computational cell as a conservative tracer subject to a constant increase with time ('aging') and mixing (Li et al., 2019) as :

$d\tau/dt = 1$,

where $\tau$ is the water age, and t is the time, with settings of initial water age set to 0 throughout the estuary and water age of 0 set on the seaward and freshwater boundaries. This method provided the temporal and spatial variation in water retention, $\tau = \tau(x, y, z, t)$. This 'water age' method is considered to have an advantage of presenting the spatial heterogeneity of water retention (Monsen et al., 2002) and suitable for long term hydrology simulations, and employed in this study to investigate the long-term evolution of water retention characteristics.

Secondly, the hydrodynamic time parameters of water flushing time (WFT) and water renewal time (WRT) were used to
investigate the changes in the mixing efficiency (ME) due to the changes in the ocean connectivity by the opening of the
artificial channel and the changes in catchment inflows. WFT is a bulk basin-wide water flushing time scale, defined as

$$WFT = V/Q$$

where V is the total estuary volume and Q is the water flux flowing out of the system. The WRT is defined as the time required
for each cell of the lagoon domain to replace the originally released conservative tracer, with settings of initial tracer
concentration set to 1 uniformly throughout the estuary, and a concentration of zero imposed on the inlets and freshwater
boundaries.
After WFT and WRT are calculated, the ME can be obtained as the ratio between WFT and WRT. ME ranges between 0 and
1. In the theoretical case of ME=1, the estuary is considered as a fully mixed system; while ME=0 indicates there is no mixing
of water masses entering the estuary with the inner waters. The ME is a useful index for investigating the mixing behaviour of
the lagoons, and also for lagoons intercomparison and classification of their ocean connection types (Umgiesser et al., 2014),
and was employed to investigate the impact of the Cut opening and catchment inflows on the estuary mixing.

**2.2.3 Model calibration, performance evaluation and sensitivity tests**

The model was calibrated with a structured hierarchical approach, similar to those described in Muleta and Nicklow (2004)
and Hipsey et al. (2020). This approach first identified the key parameters of importance to the hydrology in the current study
based on literature review and prior expert knowledge. In this stage, the key parameters were identified to be the bottom drag
coefficient, the bulk aerodynamic coefficients, and the mixing scheme options associated with the vertical turbulence model
(in this case this is parameterized through the GOTM plugin), and the bulk transfer coefficient for latent heat flux. In the
second stage, a matrix of simulations, each with pre-determined parameter vectors and model options, was assessed against
the observed salinity and temperature data at six stations within the estuary (Figure 1, at both surface and bottom levels), and
the water elevation at the centre of the Peel Inlet in year 1998, which presented a year with median rainfall and catchment
inflows. The capability of the model to reproduce the salinity stratification (magnitude of difference between the surface and
bottom salinity) created by the interaction of ocean intrusion and freshwater runoff during the wet season was also considered
in the model calibration. Based on the calibration results, a k-ε mixing scheme, a linear model for the aerodynamic coefficients
(Wu, 1982), a bulk transfer coefficient for latent heat flux of 0.0013, and the roughness scales as in Table 2 were selected.
After calibration, the model was then validated with salinity and temperature data in all simulated years (except 1970 when
the monitored data at 6 stations were not available). For each variable, we evaluated the model quantitatively against the
monitored data using three skill metrics: correlation coefficient (r), mean absolute error (MAE), and model skill score (SS).
The validation results suggest the model captured the major features of the hydrodynamic response to the external drivers of
the tide and freshwater inputs, and the model satisfactorily reproduced the salinity and temperature in both the surface and
bottom. A full description of the calibration and validation results is given in the supplementary material.

The current study focused on the impact of reduced inflow, due to drying climate and the Cut, on the estuary hydrology.
However, the perturbations of environmental factors such as air temperature, tidal elevation, and benthic vegetation could also
affect the local hydrology, and so their influence on the modelling results was explored. To evaluate the effects of these factors,
the sensitivity of the $\tau$ and salinity was assessed relative to changes in: (1) air temperature (±1 degree, representing 100 year
change of local air temperature); (2) tidal elevation (±0.15m, representing 100 year change of local tide record); and (3) bed
roughness length (±50%, representing 50% change of bed roughness). The ranges of these environmental factors were carefully
selected based on the historical records. Two years, 1990 and 1998, representing a year before the Cut-opening and another

230 year with the Cut, respectively, were selected for these model sensitivity tests. Detailed results from the sensitivity assessment
are also included in the supplementary material. In summary, the modelled salinity and τ were shown to be affected by changes
in the mean sea level and bottom roughness variations, but the effects of these factors on the results were small when compared
to that caused by the reduced flow over the past decade and the Cut-opening (shown in the next section). For example, the
maximum change in τ observed in the sensitivity test runs was 8.6 days, caused by the enhanced bottom roughness in the 1990
scenario, compared to the magnitude of 20-100 days caused by the reduced flow from 1970 to 2016 (see more details in below).
The maximum changes in the salinity observed in the sensitivity test runs was 2.8 PSU, caused by the reduction of tide level
in the Harvey Estuary, compared to the magnitude of 10-30 PSU changes in the salinity caused by the reduced flows from
1970 to 2016 (see more details in below). These results suggested the changes in the climate and the ocean connectivity are
the major drivers of the hydrology of the Peel-Harvey Estuary.

**Table 2.** Zonal characters and roughness length $z_0$ setting in the model domain.

| Zonal ID | Areas | Aquatic Vegetation Biomass (g Dry Weight/m$^2$) | $z_0$ |
|---|---|---|---|
| 1 | South/North Harvey Estuary, East Peel Inlet | 100-230 | 0.03 |
| 2 | South/North Harvey Estuary, West Peel Inlet | 50-100 | 0.02 |
| 3 | Harvey Estuary and Peel Inlet | 0-50 | 0.01 |
| 4 | Dawesville Cut | N/A | 0.003 |
| 5 | Harvey River | N/A | 0.003 |
| 6 | Murray River | N/A | 0.003 |
| 7 | Serpentine River | N/A | 0.003 |
| 8 | Coastal ocean | N/A | 0.002 |

**2.2.4 Climate change context and simulation rationale**

Historical observations of nearby precipitation and the gauged data of the major Murray River inflow have shown a decreasing
trend from 1970 to the present (Figure 2), though variability from year to year is noticeable. The average annual precipitation
has declined by 15% when comparing the period 1994-2016 relative to 1970-1993, and this led to a dramatic decrease of
annual inflow volumes, most notable in the past decade.

Years with inflow rates close to the 10-year moving average were selected for hydrologic modelling simulations to explore in
more detail the hydrologic changes occurring within these years (depicted relative to the trend in Figure 2b). Due to the concern
that the drying climate will continue into the 21st century (Silberstein et al., 2012; Smith and Power, 2014), we also undertook
model simulations to investigate potential hydrologic changes under future conditions representative of 2040 and 2060, by
considering reduced streamflow and rising sea levels. The runoff declines were based on the mean projection by Smith and
Power (2014) that suggested the total runoff to the rivers and estuaries within the south-west Western Australia region will
drop by about 0.96% per year, corresponding to the projected reduction in precipitation of 0.27% per year, on average. Sea
level rise was also included in the future scenarios, estimated from the long-term (1897 – 2000) tide gauge observations at the
Fremantle tide gauge station that shows a sea level trend of 1.50 mm/year (Kuhn et al., 2011). These estimates may be biased
due to a possible accelerated sea level rise towards the end of the 21st century (IPCC, 2007; Kuhn et al., 2011), but we highlight
these future scenarios were set up with a focus to investigate the changing hydrology into a future from the projected drying
climate trend.

For each selected year, the modelling simulation started from 1$^{st}$ September of the previous year, giving a 4-month spin-up

period, and the results from 1$^{st}$ January to the end of the selected year were used for analysis. The initial condition of water

temperature and salinity was interpolated from the field data when they were available (years 1985-2016), except the years

1970 when no field data was available and 1978 when field data at site PH31 and PH58 were missing, so the same initial

condition as for 1985 was adopted. For the future scenarios the same initial condition as in 2016 was used.

For the modelling years after 1994, when the artificial channel was constructed, we also ran "no-Cut" counter-factual scenarios,

which assumed the Dawesville Cut engineering intervention was not constructed, in order to separate the impact of the artificial

channel on hydrology relative to the "with-Cut" scenarios (Table 3).

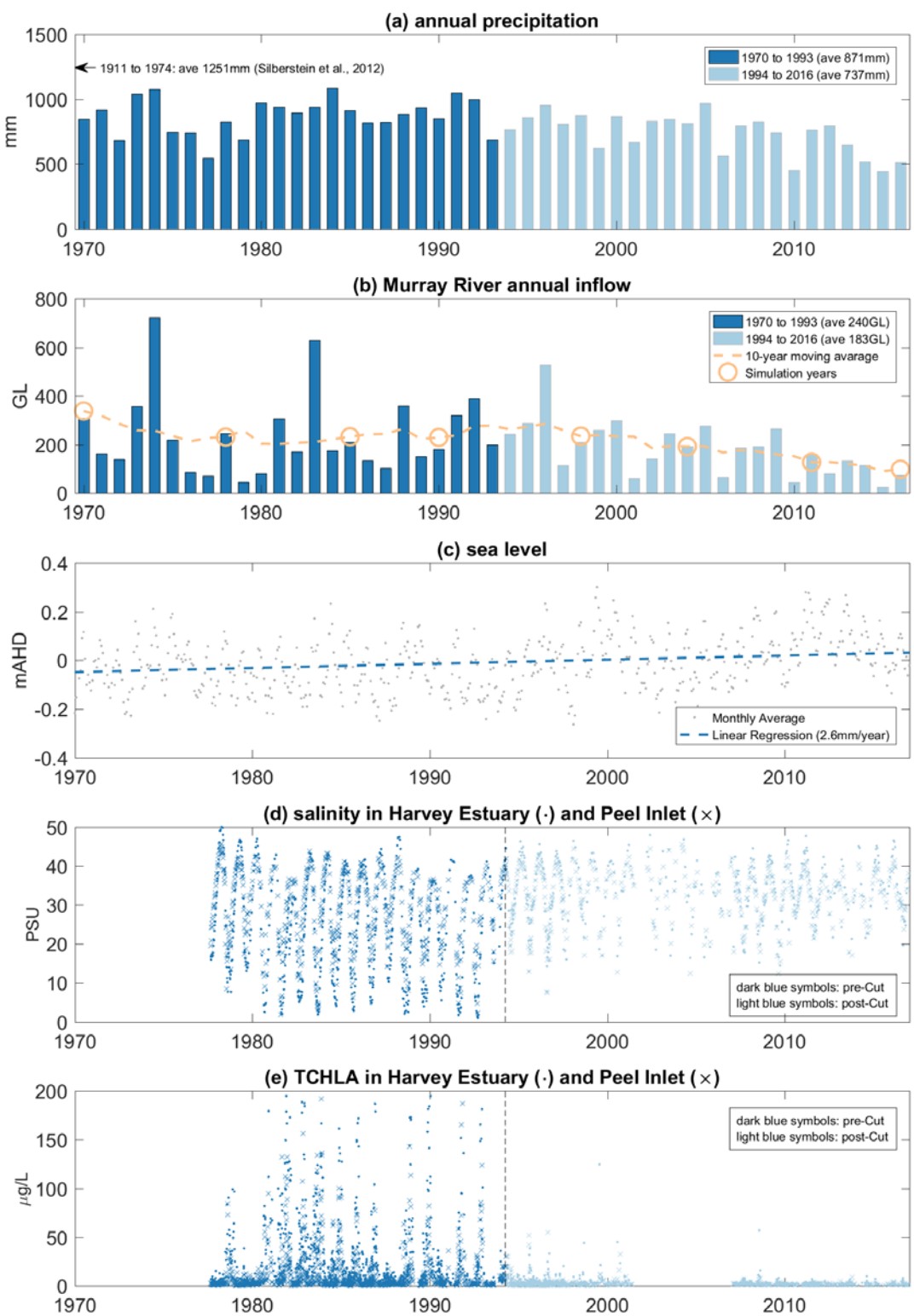

**Figure 2.** Historical record of (a) annual precipitation rate; (b) Murray River annual inflow rate; (c) monthly-average sea level at Fremantle gauge station; (d) salinity at Harvey Estuary and Peel Inlet; and (e) total chlorophyll-a (TCHLA) in Harvey Estuary and Peel Inlet since 1970 to 2016.

Gauged flow rate data for the Murray River, Serpentine River and Harvey River were applied to the hydrodynamic model whenever they are available. Gauged flow rate data for Murray River were available from 1970 to present, while for Serpentine River and Harvey River were available from 1982 to present. For the missing periods in the gauged flows and the ungauged drains, the output from the Source (eWater®) catchment modelling platform (Kelsey et al., 2011; Welsh et al., 2013), operated by the Western Australia Department of Water and Environmental Regulation, was used to estimate flows by carefully comparing the measured and modelled flow data. Groundwater inputs were previously estimated to represent only ~1% of total water inputs (Black & Rosher, 1980), and were therefore not included in the modelling simulations.

Various data sources were used to set up meteorological inputs due to the study period spanning back to 1970, when meteorological observations were not routinely available across the modelling domain at hourly frequencies. The first data source was the local Mandurah weather station located beside the natural channel of the estuary (Figure 1). This dataset provided hourly records since 2001. The hourly fields over the period 1981-2000 were obtained from regional climate model simulations for Southwest Australia at a 5 km resolution (Andrys et al., 2015; Kala et al., 2015), which were carried out using the Weather Research and Forecasting (WRF), one of the most widely used regional climate models. Andrys et al. (2015) showed that the WRF model was able to adequately simulate the climate of Southwestern Australia, and these simulations have also been used to assess the impacts of current and future climate on temperature and precipitation (Andrys et al., 2016, 2017) as well as climate indices relevant to viticulture for Southwestern Australia (Firth et al., 2017). The WRF simulations of Andrys et al. (2015) have also been benchmarked against other regional climate model simulations across the Australian continent and shown to perform well in simulating both temperature and precipitation (Di Virgilio et al., 2019) as well as heat-wave events (Hirsch et al., 2019). For the years before 1981 the weather conditions measured at the nearby Halls Head weather station (4.2 km away from the Mandurah station) were used. Though various sources of climate data were used, the wind regimes of these data sources showed similar distribution in wind magnitudes and directions (Figure 3). The winds in the Mandurah station record are relatively smaller when compared to other two sources, however, this difference may be due to the natural variation in the climate and are not expected to change the main hydrological features in the lagoon.

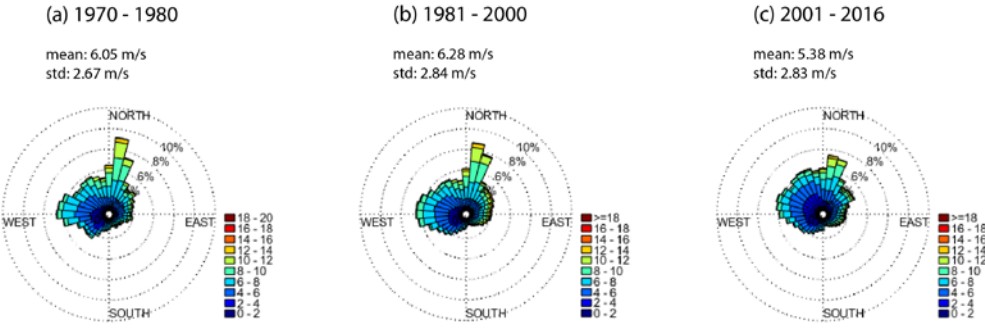

**Figure 3.** Rose plot of wind condition in years of (a) 1970-1980, obtained from the Halls Head weather station; (b) 1981-2000, obtained from the WRF weather model; and (c) 2001-2016, obtained from the Mandurah weather station.

A summary of all historical simulations and future scenarios is provided in Table 3. The total inflow into the estuary of the
chosen simulation years shows a general decrease from past to future, except for the year 1978 when the total inflow rate was
less than that in 1985 and 1990. This was due to an exceptionally low inflow rate within the Harvey River, produced from the
catchment model output, which had an effect mostly on the Harvey Estuary. We still include this year to show the historical
evolution during the past decades.

**Table 3.** Summary of simulated scenarios and their annual precipitation, catchment inflow volumes, mean sea level, and Cut-
opening information.

| Simulation Category | Simulated Year | Annual precipitation (mm) | Annual Catchment Inflow ($\times 10^6$ m$^3$) | Annual Mean Sea Level (mAHD) | Cut Opening | *Sensitivity tests* |
|---|---|---|---|---|---|---|
| *Pre-Cut years* | 1970 | 846.4 | 705.9 | -0.043 | No | |
| | 1978 | 827.5 | 591.6 | -0.024 | No | |
| | 1985 | 911.7 | 564.0 | -0.033 | No | |
| | 1990 | 849.2 | 515.8 | -0.071 | No | Yes |
| *Post-Cut years* | 1998 | 876.2 | 490.4 | -0.027 | Yes | Yes |
| | 2004 | 813.7 | 478.0 | -0.027 | Yes | |
| | 2011 | 766.0 | 378.9 | 0.156 | Yes | |
| | 2016 | 514.4 | 244.2 | 0.017 | Yes | |
| *No-Cut scenarios* | 1998 | 876.2 | 490.4 | -0.027 | No | |
| | 2004 | 813.7 | 478.0 | -0.027 | No | |
| | 2011 | 766.0 | 378.9 | 0.156 | No | |
| | 2016 | 514.4 | 244.2 | 0.017 | No | |
| *Future Scenarios* | 2040 | 481.1 | 187.9 | 0.053 | Yes | |
| | 2060 | 453.3 | 138.9 | 0.083 | Yes | |
| | 2040 | 481.1 | 187.9 | 0.053 | No | |
| | 2060 | 453.3 | 138.9 | 0.083 | No | |

## 3 Results

### 3.1 Estuary response to the change in ocean connectivity

The impact of the changes in ocean connectivity due to the construction of the artificial channel on the estuary hydrology was
first investigated by analysing the observed and modelled salinity and temperature at the centre of the two lagoons, and the
surface elevation within the Peel Inlet, in the years of 1990 (representing a 'pre-Cut' year) and 1998 (representing a 'post-Cut'
year). These two years were compared because they have similar annual precipitation and catchment inflow rates (Table 3),
and tidal forcing characteristics in terms of the annual mean sea level and tidal range (Table 4, Figure 4). Therefore, the
comparison provided a valuable insight into the impacts of the artificial channel on the estuary environment.

**Table 4.** Comparison of principal tidal constituents in year 1990 and 1998.

| Constituents | Potential Energy (%) | | Amplitude (m) | |
|---|---|---|---|---|
| | **1990** | **1998** | **1990** | **1998** |
| **K1** | 57.19 | 56.79 | 0.159 | 0.156 |
| **O1** | 30.28 | 30.05 | 0.115 | 0.114 |
| **P1** | 5.45 | 5.66 | 0.0490 | 0.0494 |
| **M2** | 3.92 | 4.31 | 0.0415 | 0.0431 |
| **S2** | 3.16 | 3.20 | 0.0373 | 0.0371 |

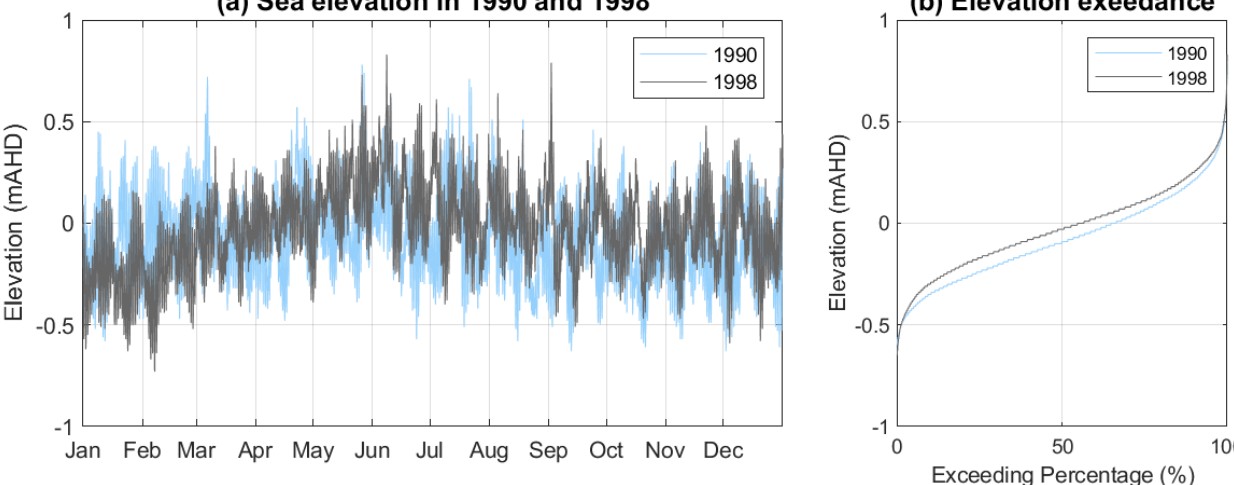

**Figure 4.** (a) Sea level changes and (b) their exceedance plot in the year 1990 and 1998.

The monitored and modelled salinity and temperature at the centres of the two lagoons demonstrate the changes in the seasonal
cycle in response to the catchment inflows and Cut opening, and the model capability in capturing these changes (Figure 5).
In summer and early autumn the flow rates were low, followed by high salinity and weak salinity stratification in the two
lagoons. In contrast, there were large inflows to the estuary in winter and early spring. The peaks of the inflows occurred in
winter (July – September), followed by a significant drop in the salinity in the estuary due to the freshwater flushing. However,
differences in the salinity response to freshwater flushing can be observed between the pre-Cut year (1990, left column of
Figure 5) and the post-Cut year (1998, right column of Figure 5). In 1990 when the estuary had limited connection without the
opening of the Cut, the salinity stratification was small in the Harvey Estuary. The salinity dropped to below 5 PSU, indicating
the hydrology of Harvey Estuary was mainly dominated by the Harvey River flushing. Whilst during 1998, with greater ocean
connection due to the opening of the Cut, stronger salinity stratification was observed in the Harvey estuary, and the minimum
salinity was lifted to over 10 PSU due to more seawater intrusion from the Cut. The water temperature also showed a clear

seasonal signal, ranging from about 10 °C in winter to 30 °C in summer. The differences in the water temperature observed in the centres of two lagoons, and between the surface and bottom waters, were small.

**Left Column: year 1990 (pre-Cut)**    **Right Column: year 1998 (post-Cut)**

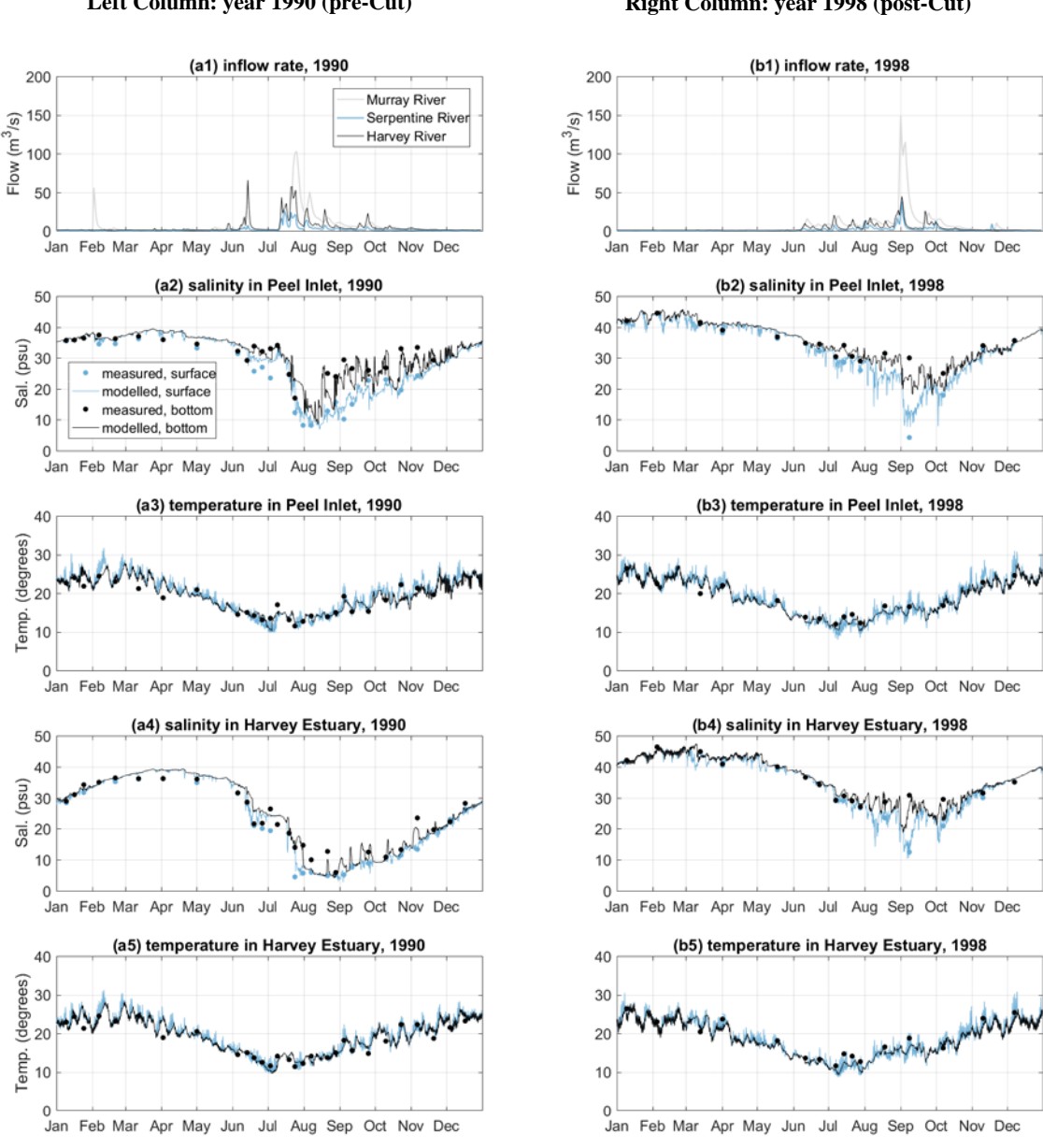

**Figure 5.** Annual variation in 1990 (left column, a) and 1998 (right column, b) of (1) inflow rate of the three main rivers; (2) monitored and modelled surface and bottom salinity at the centre of Peel Inlet (site PH7 at Figure 1); (3) monitored and modelled surface and bottom water temperature at the centre of Peel Inlet; (4) monitored and modelled surface and bottom salinity at the centre of Harvey Estuary (site PH1 at Figure 1); (3) monitored and modelled surface and bottom water temperature at the centre of Harvey Estuary;

The opening of the Cut also affected the surface elevations of the estuary (Figure 6). The estuary surface elevation in 1998 had a much wider range of -0.6 m to 0.8 m compared to that in 1990 of -0.4 m to 0.4 m, indicating an enlarged tidal-prism and higher magnitude of water exchange with the ocean due to the opening of the Cut.

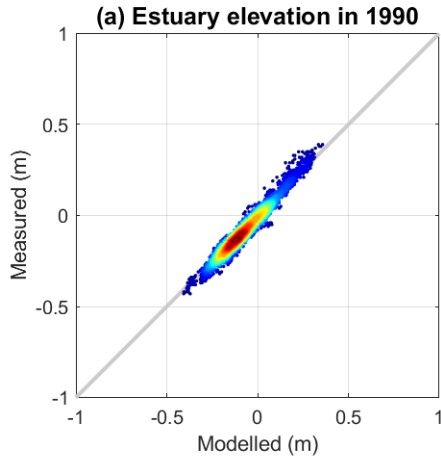 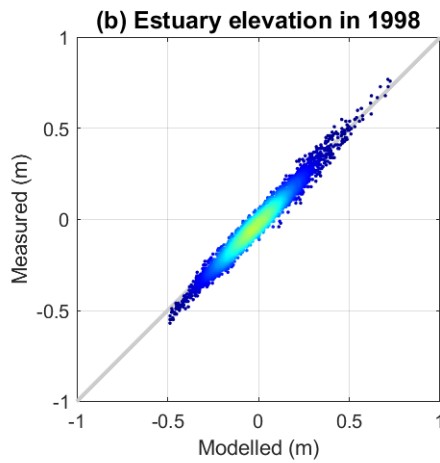

**Figure 6.** Modelled vs. measured surface elevation in the centre of Peel Inlet in (a) 1990 ($r$=0.9795), and (b) 1998 ($r$=0.9841). The grey line indicates the 1:1 ratio; the colour from blue to red indicates the data density from minimum to maximum.

The impacts of the changes in ocean connectivity and catchment flows on the estuary mixing is then explored with results of WRT, WFT, and ME in four short theoretical scenarios (constructed to represent a matrix of open/close of the artificial channel and wet/dry catchment inflows):

- Scenario 1: Cut open, with inflow condition in February 1998 (dry month with mean inflow rate of 5.60 m³/s);
- Scenario 2: Cut open, with inflow condition in August 1998 (wet month with mean inflow rate of 40.23 m³/s);
- Scenario 3: Cut closed, with inflow condition in February 1998 (dry month with mean inflow rate of 5.60 m³/s);
- Scenario 4: Cut closed, with inflow condition in August 1998 (wet month with mean inflow rate of 40.23 m³/s);

The results of fluxes at the inlets, fraction of lagoon water volume exchanged daily with the sea (FVE), average WRT, WFT, and ME are summarised in Table 5. The presented results highlight the hydrodynamic variability with the changes in catchment runoff and ocean connectivity. The bulk WRT ranges from 154.80 days with small catchment flows in the dry seasons and with only the natural channel (scenario 3), to 13.92 days with higher catchment flows in wet seasons and with both the natural and artificial channels (scenario 2). The exchange flux through the inlets between the lagoon and ocean increased by ~3 times due to the opening of the artificial channel. The wet months had higher ME compared to the dry months. However, it is interesting that the opening of the artificial channel, though largely reduced the water retention time scales, resulted in similar ME in the dry month and less ME in the wet month.

**Table 5.** Model simulation results for the average water flux through the channels (Flux), fraction of lagoon volume exchanged daily with the ocean (FVE), WRT, WFT, and ME in selected scenarios.

| Scenario | Opening of Dawesville Cut | River Runoff (m³ s⁻¹) | Flux (m³ s⁻¹) | FVE | WRT | WFT | ME |
|---|---|---|---|---|---|---|---|
| 1 | Yes | 5.5973 | 437.6398 | 0.2016 | 31.6113 | 4.9593 | 0.1569 |
| 2 | Yes | 40.2253 | 601.8528 | 0.2773 | 13.9174 | 3.6061 | 0.2591 |

| 3 | No | 5.5973 | 98.6111 | 0.0454 | 154.7988 | 22.0093 | 0.1422 |
| 4 | No | 40.2253 | 170.0371 | 0.0783 | 34.2624 | 12.7641 | 0.3725 |

The spatial distribution of WRT corresponding to the changes in the inflow condition and the opening of the artificial channel is shown in Figure 7. These maps clearly identify areas where waters are either well flushed or poorly flushed, and show the Peel-Harvey system exhibiting a highly heterogeneous spatial distribution of the WRT. In all scenarios, WRT is mainly dependent on the relative distance from the inlets and on the presence of channel. The areas connected to these channels are directly influenced by the sea and consequently their water renewal times are lower. In the wet season, the river runoff also plays a role in determining the water renewal heterogeneity. The south Harvey Estuary is shown to have the highest WRT than other parts of the lagoon, indicating the poorly flushing in this area.

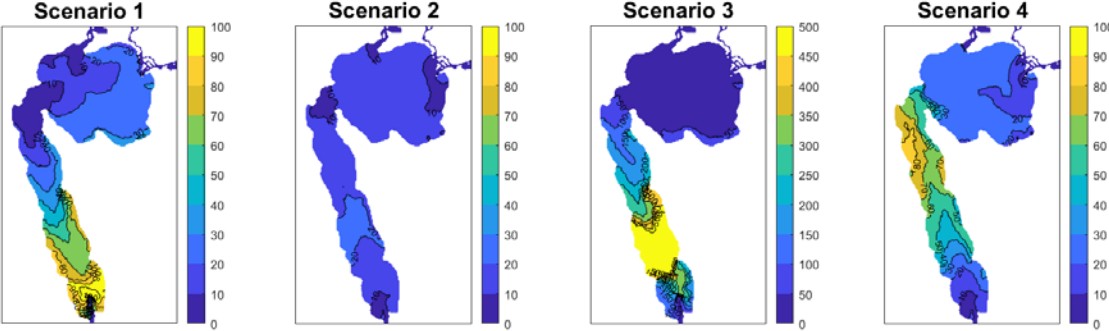

**Figure 7.** Spatial distribution of modelled WRT in 4 selected scenarios.

**3.2 Long term response of water retention to climate change and Cut-opening**

Water retention is highly dynamic depending on seasonal flows, tidal conditions, and in different regions of the estuary. The evolution of water age, $\tau$, over time has shown a general increase from1970 to the present, superimposed on the effect of the Cut, however, with some considerable variation across the two lagoons (Figure 8). Firstly, the wet season (winter and spring) was more sensitive to the changes in the drying climate. In the "no-Cut" scenarios (assuming the artificial channel was not constructed), it was predicted that $\tau$ would have increased in the Peel Inlet from about 50 days in 1970 to nearly double in 2016, and increased from approximately 50 days in 1970 to nearly 150 days in 2016 in the Harvey Estuary, solely due to the drying climate trend. In contrast, the dry season (summer and autumn) conditions did not show significant changes over time in most parts of the estuary, except in the south Harvey Estuary, which is furthest from the channels. The opening of the Cut had a prominent effect by reducing $\tau$ by about 20-45 days in the Peel Inlet, and more profoundly by 50-100 days in the Harvey Estuary. Yet the drying climate effect on the water age has largely cancelled out the flushing effect by the Cut in some regions. The increases in $\tau$ from 1998 to 2016, due to reduced inflows, are of the same magnitude as the level of reduction caused by the Cut opening. For example, the Cut opening reduced the $\tau$ by 28 days in the west Peel Inlet in 1998, yet the $\tau$ increased by 27 days from 1998 to 2016 due to the reduced flows. Lastly, the Harvey Estuary was most influenced by the climate changes and the Cut opening. North Harvey Estuary, directly adjacent to the Cut, was most impacted by the Cut opening, and the $\tau$ was reduced by more than 110 days. The south Harvey Estuary, which is furthest from both the channels, was more sensitive to climate change, showing the greatest variation over the most recent decade. The projected climate is expected to increase the $\tau$ further in the Harvey Estuary in spring, but a relatively smaller impact at other sites and seasons.

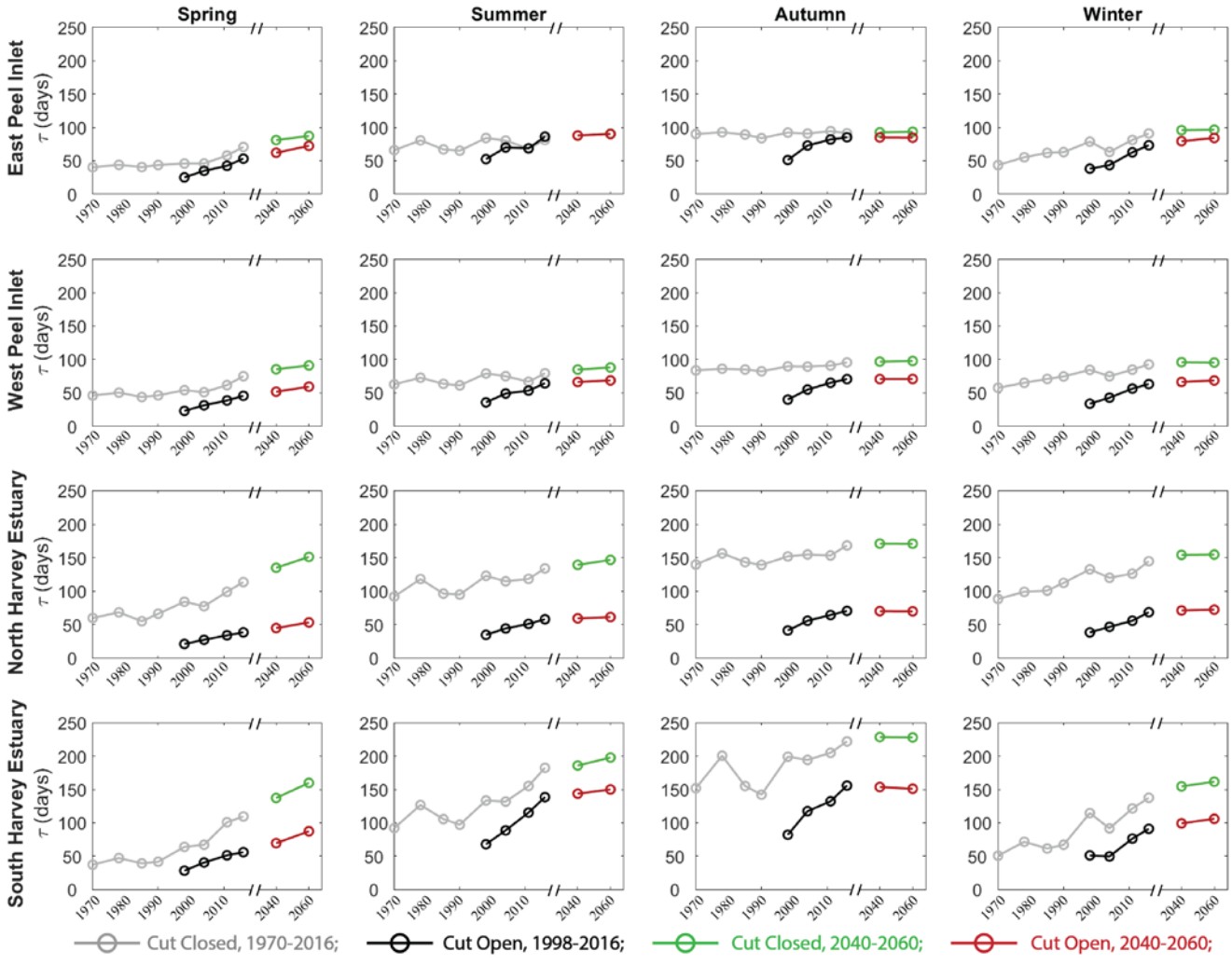

**Figure 8.** Mean water retention time, $\tau$, in east Peel Inlet, west Peel Inlet, north Harvey Estuary, and south Harvey Estuary (see Figure 1 extents) in simulated years and future scenarios. The data were categorized into four seasons: spring (September, October, November), summer (December, January, February), autumn (March, April, May), and winter (June, July, August).

The spatial difference in water age is further illustrated in Figure 9, which shows a plan-view of the seasonally-averaged water ages in year 1990 and 1998. The spatial distribution pattern of water age is similar to the one of WRT (Figure 7), showing that, the water age in the areas adjacent to the Cut entry point has been largely reduced by the Cut-opening, yet the south Harvey Lagoon and some parts of the east Peel Inlet still showed high water retention.

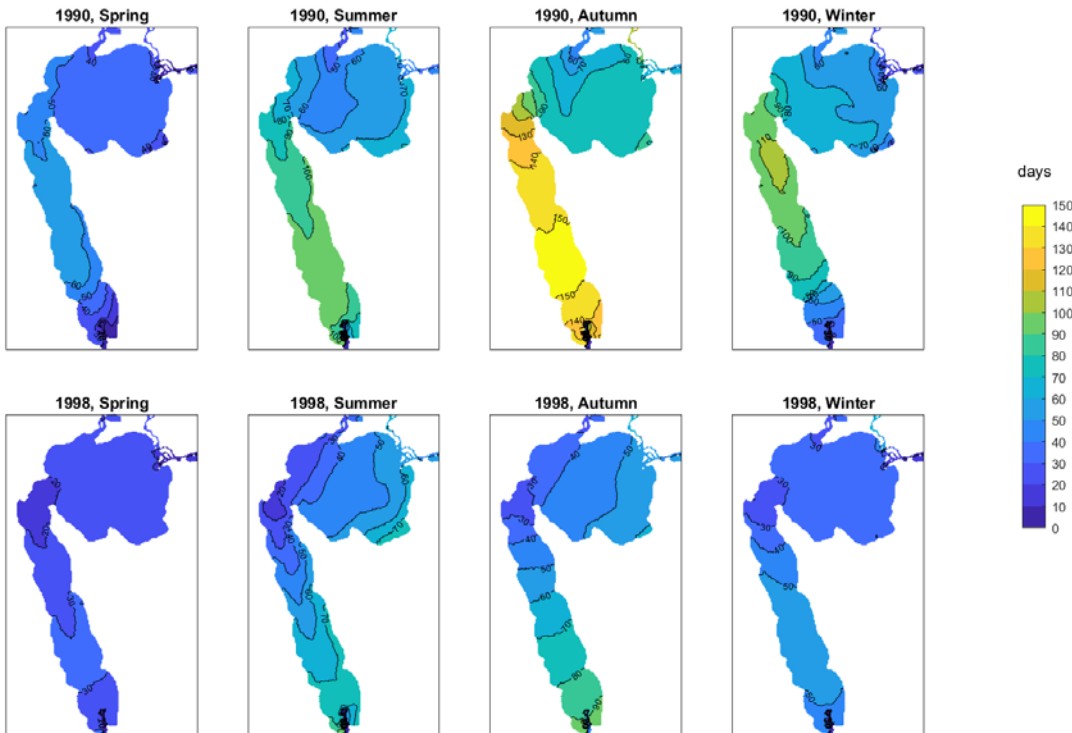

**Figure 9.** Spatial distribution of season-averaged water age in 1990 (top panels) and 1998 (bottom panels).

### 3.3 Long Term Responses of salinity and stratification to climate change and the Cut-opening

Similar to $\tau$, the changes in salinity in response to the drying climate showed large variability in space and time, and the impact of the Cut-opening acted to increase salinity in the wet season, but reduced hypersalinity risks (>40 PSU) in the dry season (Figure 10). In the "no-Cut" scenarios, the mean salinity during the wet season increased from <20 PSU in 1970 to over 30 PSU in 2016. During the dry season, the changes in salinity were relatively smaller over time. The Cut-opening could increase or decrease the salinity in the estuary, depending on the salinity within the estuary at the time, compared to the ocean salinity of approximately 36 PSU. If the estuary salinity was lower than the ocean salinity, the Cut-opening tended to increase the salinity level, and vice versa. For example, the salinity in the north Harvey Estuary increased from 17.5 PSU to 28.3 PSU in the spring of 1998 by the Cut-opening, yet reduced from 51.8 to 39.8 PSU by the opening of the Cut in the autumn. The Cut-opening has a relatively smaller influence on the salinity of Peel Inlet, which is connected with the ocean via not only the Cut but also the Mandurah Channel. The projected climate is expected to slightly increase the salinity in the Peel Inlet and Harvey Estuary, mostly in the winter and spring periods.

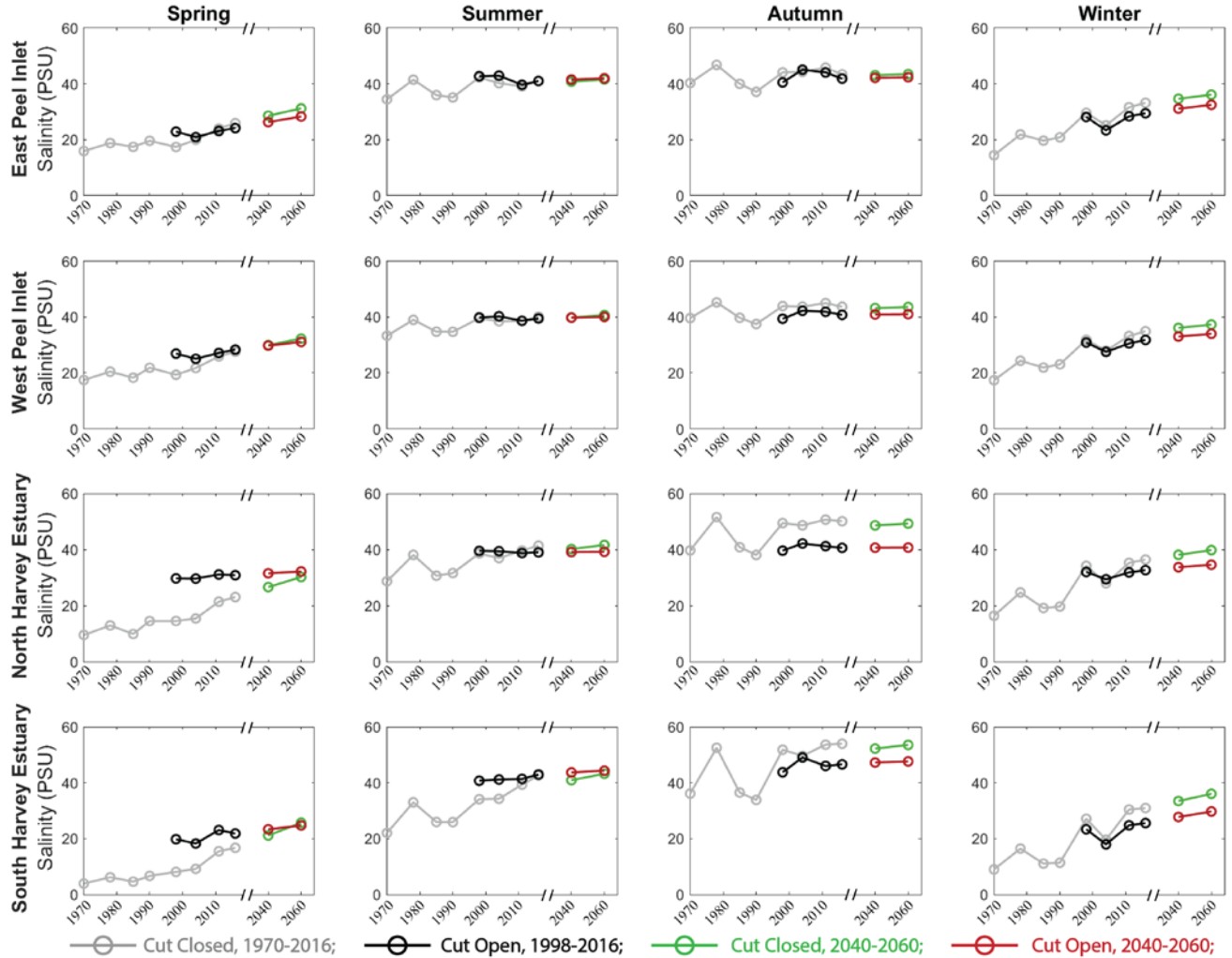

**Figure 10.** Changes of mean salinity in PHE in simulated years and future scenarios. As for Figure 8 the changes are categorized into four zones and four seasons.

Hypersalinity was often observed in the summer and autumn seasons in the Peel Inlet from both the 'with-Cut' and 'no-Cut' scenarios. The Harvey Estuary shows an increasing salinity with the drying climate in summer and becomes hypersaline after 2011. High salinity with values over 50 PSU was observed in autumn in South Harvey Estuary in the 'no-Cut' scenarios, whilst the Cut-opening reduced the hypersalinity risks in autumn in the Harvey Estuary. The relationships of the hypersalinity and the catchment inflows are further investigated with monitoring data at six regular monitoring sites (Figure 11), which highlights the maximum salinity recorded in autumn has increased with reduced inflows, especially in the period before the Cut-opening. Opening of the Cut reduced the maximum salinity at the sites near the Cut (site PH2 and PH58) under an annual flow threshold of about $1000 \times 10^6$ m$^3$/year. The hypersalinity risk increases with distance from the channels, especially in site PH31 in the south Harvey Estuary where salinity >45 PSU was often observed after the Cut-opening. The maximum salinity can also be affected by other factors, such as unseasonal rainfall events in summer, which brought down the maximum salinity measured in March (Figure 11). However, it can be concluded that the hypersalinity risks have increased in response to the catchment drying trend, and the Cut-opening has reduced the sensitivity of maximum salinity to the changes in inflow rates.

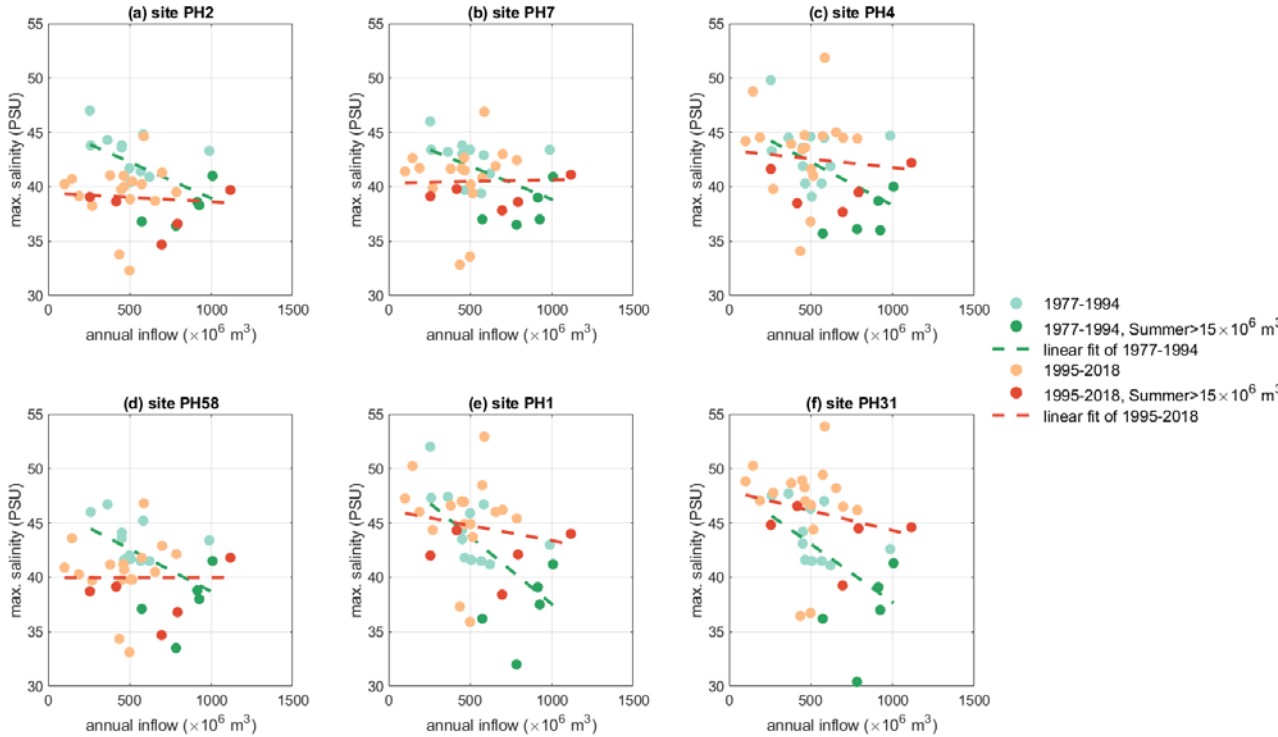

**Figure 11.** Maximum salinity recorded in March/April and the annual inflow in the hydrologic year (March to March) at 6
monitoring sites (see the site locations in Figure 1). The darker symbols indicate the years with accidental summer rainfall
events, during which the catchment inflows higher than $15 \times 10^6$ m$^3$.

The magnitude of salinity stratification (salinity difference between the bottom and surface water) in winter and spring has

shown a declining trend with the drying climate, while the variations were small in summer and autumn (Figure 12). The

opening of the Cut has enhanced the rate of ocean water intrusion, which creates stronger salt-stratification during the wet

season when it interacts with the freshwater inflows. The salt stratification was reduced to mostly < 2 PSU in the 2060

projection scenario, indicating a weaker salt-stratification due to the reduced freshwater inflows and sea level rise.

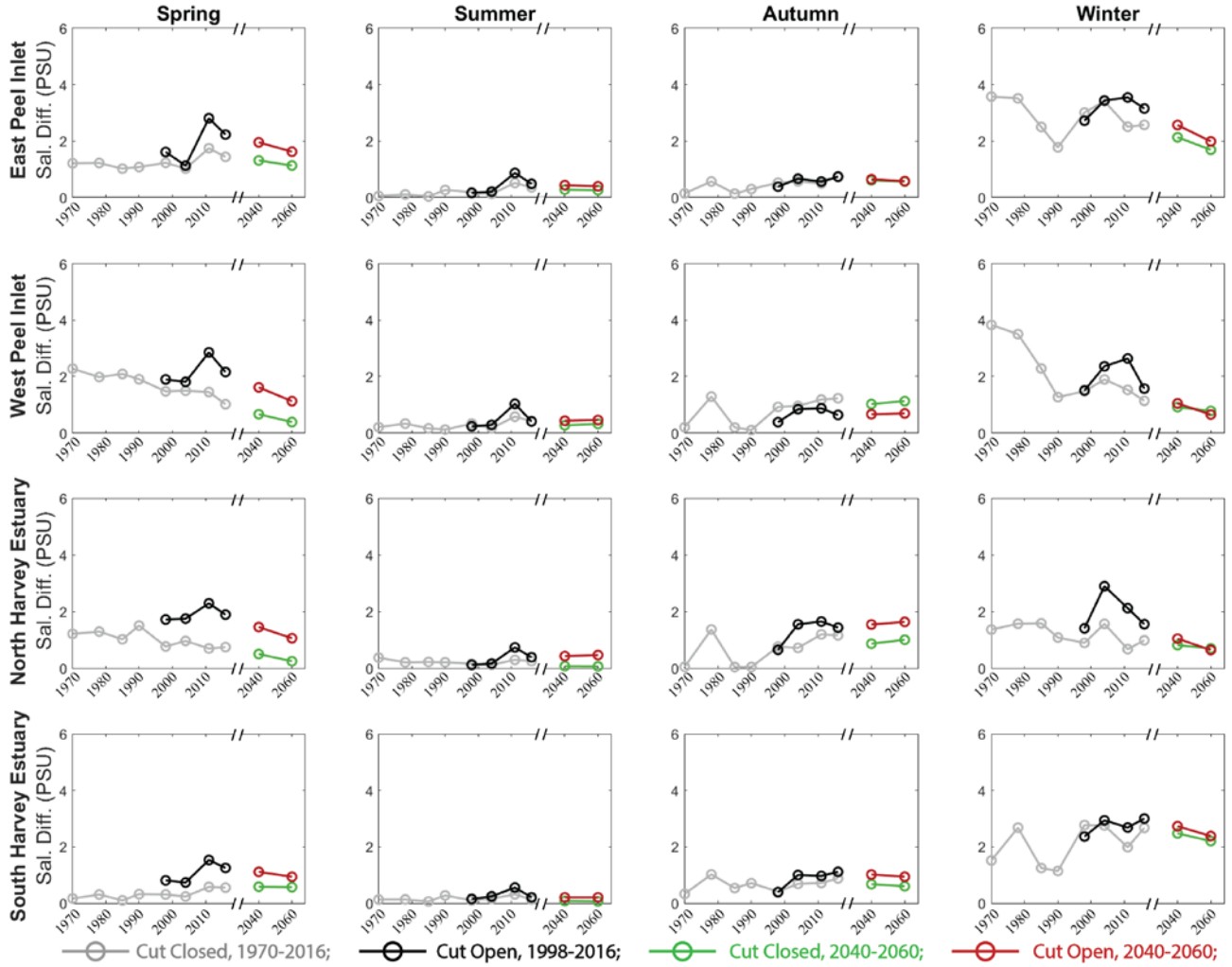

**Figure 12.** Changes of mean salinity difference between surface and bottom waters in PHE in the simulated years and future scenarios. As for Figure 8 the changes are categorized into four zones and four seasons.

## 4 Discussion

### 4.1 Changes in flushing and mixing with increasing ocean connectivity

According to Kierve and Magill (1989), coastal lagoons can be conveniently subdivided into choked, restricted, and leaky systems based on the water exchange between the lagoon and the ocean. Umgiesser et al. (2014) compared 10 Mediterranean lagoons and classified the lagoon types based on the WRT and the fraction of lagoon water volume exchanged daily with the open sea. Based on the numbers in their comparison, the Peel-Harvey Estuary can be classified as a restricted type before the opening of the artificial channel, and as a moderately leaky type after the opening of the artificial channel.

The reduction in the catchment runoff led to a smaller fraction of lagoon water volume exchanged daily with the open sea, but the magnitude is much smaller than that introduced by the opening of the artificial channel. However, the catchment runoff is shown to increase the mixing efficiency. During the dry season, the Harvey Estuary lagoon, especially the southern area, received relatively lower rates of ocean flushing that effectively lowered the ME. Higher values of ME can be found in wet seasons that enhanced the mixing of the lagoon. The increased ocean connectivity by the opening of the artificial channel,

though enhances exchange fluxes, is shown to lower the ME. This is similar to the findings in Umgiesser et al. (2014) where they found the exchanges with the open seas were low in more restricted type lagoons such that the wind has more time to mix the basins well.

The impact of the artificial channel on the water transportation was further explored by the residual currents (Figure 13), calculated as the mean currents over the period of selected scenarios described above. The results suggested that the opening of the Dawesville Cut had a strong impact on the residual currents. In the scenarios with the Cut open, strong residual currents around the Dawesville Cut are observed, and during the wet months the catchment runoff from the Serpentine River and Murray River flow to either the Mandurah channel or the Dawesville Cut. Whilst in the scenarios with Cut closed, the surface residual currents in the Harvey Estuary were mostly moving northward, and during the wet months the catchment runoff from the Serpentine River and Murray River formed a 'short cut' to the Mandurah channel via the Peel Inlet, with the west Peel Inlet received relatively less flushing. The results also indicated the surface residual current speeds in the shallow water of the basins, such as the south-east area of the Peel Inlet, were relatively lower than that in the deeper water, indicating less flushing in these areas that is coincident with the spatial distribution of WRT (Figure 7) and water age (Figure 9).

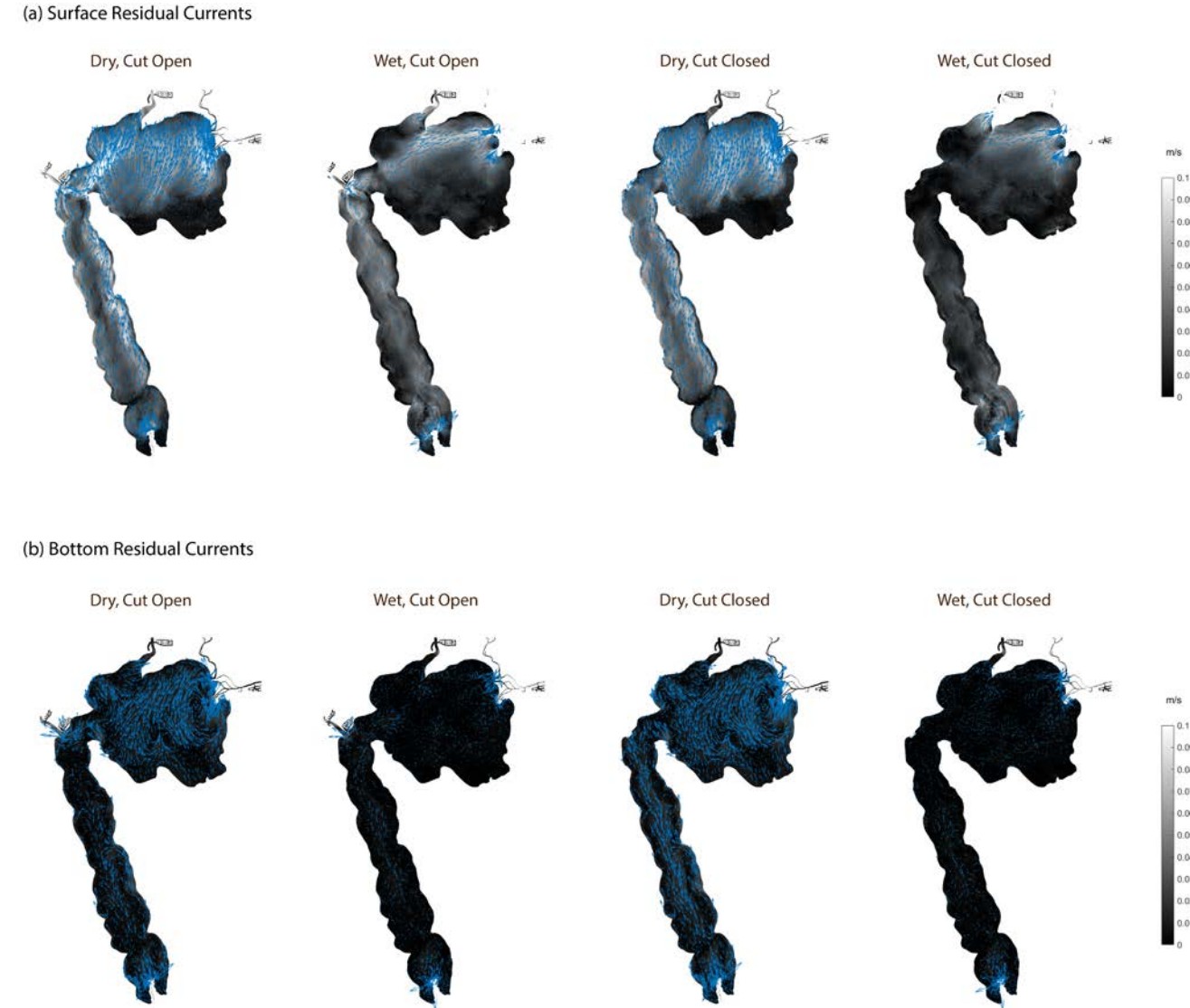

**Figure 13.** Plain views of residual currents in the selected scenarios. The black-white colour gradient indicates the total current speed; the blue arrows indicate the current vectors.

**4.2 Individual and combined impacts of the drying climate and artificial channel on the lagoon hydrology evolution**

The results of the long-term changes in the water age and salinity clearly showed the hydrology in PHE was profoundly changed corresponding to the reduced precipitation and catchment inflow as well as to the opening of the artificial channel, although other factors such as changes in air temperature, sea level rise, and benthic roughness also affected the hydrology over much smaller scales. The results have highlighted magnitudes of hydrologic changes introduced by the drying climate, and the complexity of the interacting impacts from climate and the artificial channel in time and space. Firstly, the artificial channel successfully improved the estuary flushing by reducing average water age by 20-110 days; while in contrast the reduced precipitation and inflow had a gradual opposite effect on the water age, and during the wet season this has almost counteracted the reduction brought about by the channel. Secondly, the drying climate caused an increase in the salinity by 10-30 PSU; whilst the artificial channel increased the salinity during the wet season, it has reduced the likelihood of hypersalinity (>40 PSU) during the dry season in some areas.

The climate factor had not been considered in previous reports evaluating or predicting the consequence of the Cut-opening when it was originally designed (Lord, 1998; Manda et al., 2014; Prestrelo & Monteiro-Neto, 2016), as the focus was on the flushing benefit to reduce the accumulation of nutrients and algal biomass. The findings from this study suggest that climate change has been taking effect over the period when the Cut was implemented, and from the view point of particular metrics, it is now over-taking the effect of the Cut in its significance. The lessons from this case-study highlight the need to look more broadly at environmental impacts when designing or operating large-scale engineering projects on coastal lagoons, due to the potential for long-term non-stationarity in contributing river flows.

Of relevance to management, the impacts also varied spatially in this large lagoon. The water age and salinity have showed distinct responses to the climate change and Cut-opening with various connection with the rivers and ocean (Figure 8 and 10). The southern Harvey Estuary, which has the least connection with the ocean through the natural channel, is the most sensitive to climate change and the opening of the artificial channel. The bulk flushing time also showed significant reduction corresponding to the Cut-opening, however, it was less sensitive to the drying climate. The results of water age distribution indicated that incomplete mixing had led to area-specific retention of water. In this case the concept of bulk flushing time therefore needs to be used with caution in such a large choked-type lagoon, because it only gives an average estimation of water retention for the whole estuary and fails to consider the strong gradients in lagoon hydrodynamics. Understanding these patterns can be important to help understand local effects on lagoon ecology (e.g. crab larval recruitment) and processes related to nutrient deposition and retention within the sediment.

Aside from changes in flushing and the mean salinity fields within the lagoon, the changes in the climate and ocean flushing also altered the hydrology in the tidal reaches of the rivers connecting to the PHE. The annual variability of salinity along the rivers (Figure 14) indicated there is an increasing risk of hypersalinity in the Serpentine River (connecting to the PHE from the north) and an upward movement of the salt-wedge in the Murray River (the major inflow connecting PHE from the east). For example, the mean salinity at the Serpentine River mouth was about 20 PSU in 1970, then increased to 24 PSU in 1998 and projected to increase to over 30 PSU in 2060. In the upstream areas of the Serpentine River, the mean salinity increased from about 15 PSU in 1970 to near 35 PSU in 2060. While there is less hypersalinity risk in the Murray River due to larger volumes of freshwater flushing, there is also a trend of increasing salinity along the river with the drying climate. The

550 differences between the Cut-closed and Cut-open scenarios in year 1998 are much smaller than those caused by the drying
climate, which indicates that the drying climate is the major cause of the salinity changes in the rivers.

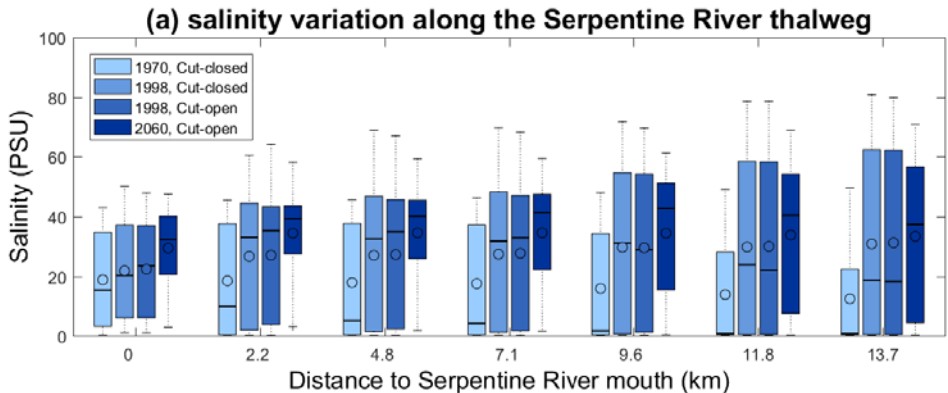

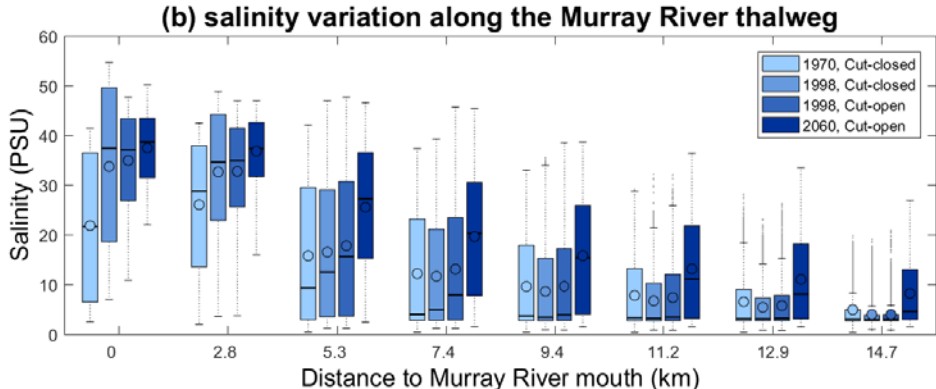

**Figure 14.** Longitudinal gradient in annual salinity variability in four selected scenarios (1970, 1998 without the Cut opening,
1998 with the Cut opening, and a future scenario 2060 with assumptions of reduced flow and sea level rise) moving upstream
along the (a) Serpentine River and (b) Murray River.

**4.3 Potential impacts from the morphological change and catchment development on the Peel-Harvey hydrology**

In the current study, we assumed that the morphological change over the study period (except through the construction of the
Dawesville Cut) were not significant to the overall hydrology. The morphology data we used for the model was the latest
morphology dataset from the Western Australia Department of Water, obtained in 2016 (integrated DEM at 2m resolution).
There was no historical topography data available for each of the selected simulated years, therefore the 2016 morphology was
applied to the study period. The changes in the morphology during this long term period could potentially affect the hydrology
and the interpretation of the results.
The estuary morphology over the study period may have been modified by: (1) changes to the net sedimentation of particles;
and (2) dredging activities related to estuary management such as marina and navigation channel developments. The net
sedimentation rates in the Peel-Harvey Estuary had been investigated by a few early studies. Gabrielson and Lukatelich (1985)
estimated a net sedimentation rate of about 0.4-1.5 mm/year in the Peel Inlet and 2.9-6.7 mm/year in the Harvey Estuary;
Hodgkin et al. (1980) estimated an overall rate of sediment deposition of 0.3 mm/year. Assuming the rate is constant, the
maximum total sediment deposition is about 75 mm in the Peel Inlet and 335 mm in the Harvey Estuary, over a period of 50

years. Note these rates were estimated in the time before the Dawesville Cut was constructed. After the year of 1994 when the Dawesville Cut was constructed, the sediment deposition in the system, especially in the Harvey lagoon, were expected to decrease due to higher tidal flushing. The reduced flow rates over the course of the study period would also lead to a reduction in sediment loading. As we illustrated in our sensitivity tests, a change in the tidal elevation of ±0.15m, which could be theoretically equivalent to the change in the depth of the estuary, was predicted to have caused a relatively small change to the hydrology compared to that introduced by the opening of the artificial channel and the reduced flow. Therefore we assumed the impact of sediment deposition on the morphology was small and unlikely change the main conclusions from the current study.

The development of canal estates and navigation channels would have further changed the local morphology, but is expected to only slightly modify the estuary hydrology at the regional scale. For example, The Yunderup navigation channel, located at the east side of the Peel Inlet, is one of the more significant dredging projects in the Peel-Harvey Estuary in the past decades. The Yunderup channel has a length of ~3km (mostly in the canal estate and shallow water areas) and a width of ~50m. The total area of this channel is ~0.015 km2, which is negligible when compared to the area of the east Peel Inlet of 33.5 km$^2$. We therefore assume the changes brought about by the local dredging activities are negligible in analysing the estuary hydrology when looking at the average properties over the regions.

Another concern is that the catchment runoff will not only be affected by the effects of reduced rainfall, but also due to the land-use change, urban development, and water diversion. The impacts from the catchment development on the flow conditions has been extensively discussed in the Peel-Harvey catchment modelling report (Kelsey, 2011), which showed different catchment developments had a combined effect on the flows. For example, the land clearing is expected to increase the streamflow, while local drainage changes have led to water diversions and reductions of the inflows. The catchment development was estimated to lead to a net increase of annual flow of about $45 \times 10^6$ m$^3$ due to the land clearing, water supply and irrigation dams, and the drain activity (Kinhill Engineers, 1988; Kelsey, 2011). The net change in the streamflow is relatively small when compared to the reduced annual flows entering the Peel-Harvey Estuary from 1970 ($846.4 \times 10^6$ m$^3$) to 2016 ($514.4 \times 10^6$ m$^3$). Regarding the potential for changes in water diversion, there are 15 dams in the Peel-Harvey catchment and the total catchment area to be dammed is 1,283 km2, which is about 12% of the total Peel-Harvey catchment (10,671 km2) (Kelsey, 2011; Hennig et al., in prep). Most of the dams were completed in the time before 1970s when our study period started. The latest catchment modelling results (Hennig et al., in prep) showed that the average annual flow in the years of 2006-2015 from unrestricted catchments was $369 \times 10^6$ m$^3$/year while the flow from dammed catchments was $36 \times 10^6$ m$^3$/year, which would amount to an additional 10% increase in annual flow if these dams didn't exist. It is understood that no dams in the catchment have planned environmental water releases and it is expected that any water releases would either be a small proportion of flow. In the hydrologic models we have used the gauged flow data combined with the catchment model outputs, therefore the changes of the flow rates due to the catchment development have been accounted for in the model settings and our analysis. Therefore, we expect the reduced inflow was predominantly a result of the drying climate, and a key factor of reducing the nutrient loads from catchment that subsequently reduced the nutrient concentrations in the estuary.

**4.4 Uncertainty of future hydrology**

This study has investigated the hydrologic changes under projected future drying climate, however, the drying climate was idealised based on trends from a combination of climate models (Smith and Power, 2014) and the applied annual perturbations used to generate the future climate remains the subject of uncertainty. Our future climate projections for weather and flow change were based on the average trend reported from more detailed studies using an ensemble of climate models (Silberstein et al., 2012; Smith and Power, 2014). The Peel-Harvey region has experienced a widely reported decline in rainfall over the

last several decades (CSIRO & BoM 2007; IPCC 2007; CSIRO 2009; Hope & Ganter 2010). The trend in rainfall decline is expected to continue, based on the climate projections from general circulation models (GCMs) results (CSIRO 2009; Smith and Power, 2014). Given the nature of our research questions was to extrapolate the mean trend that we reported from the hind-cast simulations, we focus the future scenarios on the changes of hydrology under the projected average reduction in the flow from the ensemble models (Smith and Power, 2014), with an assumed mean rate of sea level rise (Kuhn et al., 2011), to highlight the general trend and allow for prioritization of adaptation strategies such as environmental water allocation policies. This approach is somewhat simplistic in that it assumes no seasonal change in hydrologic trends, and there has been recent evidence that increasing summer floods are occurring and the winter peak flows are decreasing as a fraction of the annual total (McFarlane et al., 2020). As shown in Cloern et al. (2016), the hydrology of lagoons has been changing at a faster pace in the past decade from a combination of human activity and climate variability. The sea level of the ocean adjacent to the PHE has been rising at faster speeds in the past decades (Kuhn et al., 2011). The PHE catchment is also undergoing fast development due to the increasing population and agricultural expansion (Kelsey et al., 2011). Intensification of human activities, such as water consumption and diversion, will further affect the lagoon's hydrology and associated ecosystem, but how these factors will change in future remains unclear. Therefore, our results related to the future prediction are simply to indicate the possible changes of hydrology under the projected drying climate in order to highlight the general trend and allow for prioritisation of adaptation strategies such as environmental water allocation policies. Continuous monitoring on the hydrology and water quality of the lagoon and its catchment must therefore be prioritised to closely observe further hydrologic change in order to provide prompt actions for management.

**4.5 Applications for estuary ecosystem management**

The Cut had an obvious and dramatic effect on increasing the export of nutrients that would have otherwise been retained (Figure 15). Since the Cut opening in 1994 the main monitoring stations have shown the total nitrogen (TN) concentration as being stable around 0.5mg/L and the total phosphorus (TP) concentration has declined from 0.05 to 0.02 mg/L over time. Importantly, the increasing rate of exchange has made the estuary concentration of nutrients less sensitive to the inflow load (as demonstrated by the reduction in slope of Figure 15c and 15d). The results have also revealed an increase in τ associated with the drying climate that has eroded some of the benefits associated with increased flushing following the construction of the Cut, and further reductions in flows will cause less flushing and will likely lead to a tendency for increasing nutrient accumulation over time.

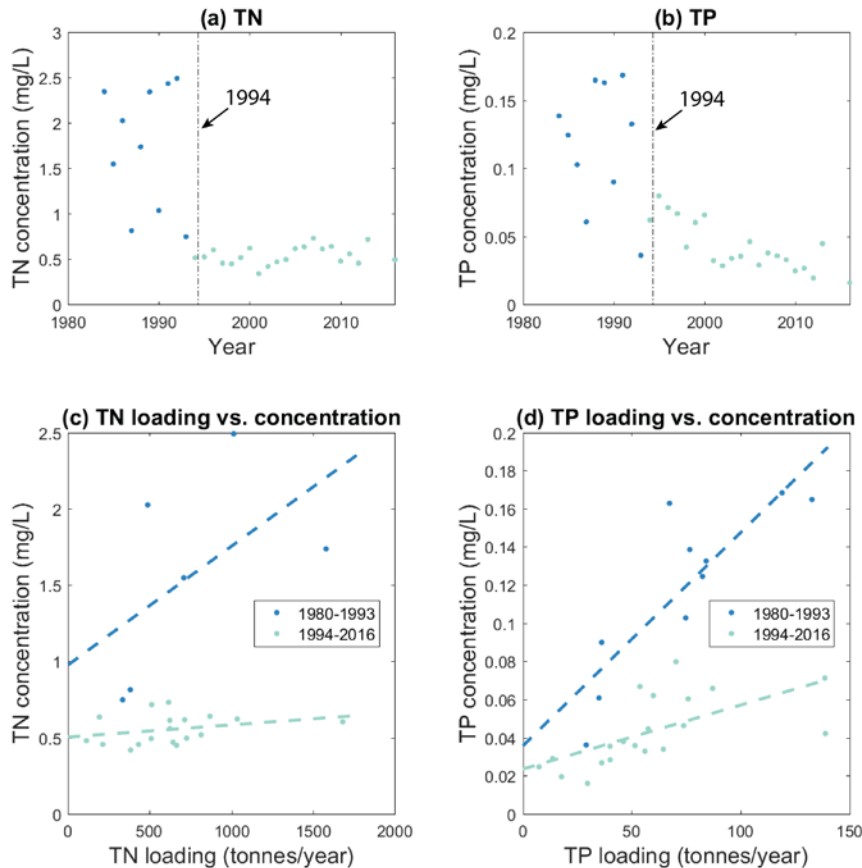

**Figure 15.** Changes in the mean nutrient concentration of (a) TN and (b) TP in the Peel-Harvey Estuary (based on the average of the 6 main monitoring stations), and their relationship with the total annual nutrient loading (c and d).

The hydrologic changes led not only to changes in the nutrient concentrations but also the mean salinity, with potential ramifications for the ecological community. In particular, the phytoplankton biomass dropped dramatically since the Cut opening (Figure 2) due to the improvement of ocean connectivity and flushing, but also due to a less desirable salinity regime in summertime for the toxic cyanobacteria *Nodularia spumigena* that plagued the Harvey Estuary before the Cut opening. Field observations also showed that the biomass of macroalgae has decreased in the Peel Inlet, while it has increased in the Harvey Estuary since the Cut opening, which potentially reflects the reduced nutrient concentrations, increased salinity and greater light availability (Pedretti et al., 2011). The biomass of some benthic macroinvertebrates, such as the blue swimmer crabs (*Portunus armatus*) and the Western king prawn (*Penaeus latisulcatus*) also showed an increase with the Cut opening and the reduced flow in recent years (Bradby, 1997; Johnston et al., 2014). Nonetheless, several water quality concerns remain present, and problematic areas of poor quality hyper-sulfidic sediments have been identified (Kraal et al. 2013; Hallett et al., 2019), in addition to recurrent reports of harmful algal blooms and fish-kill events within the inland reaches (Valesini et al., 2019). The changes in water quality and the biological communities are anticipated to continue to evolve as the projected drying climate will further lead to areas of poor flushing and high salinity.

## 5 Conclusions and outlook

This study has sought to analyse the hydrologic changes in the Peel-Harvey Estuary to a range of drivers, and focused on the effects of the recent climate change trend on the hydrologic evolution in the Peel-Harvey Estuary, relative to the changes brought about by construction of the Dawesville Cut. Our results suggested the climate change in the past decades has a remarkable effect on the hydrology with the same magnitude as that caused by the opening of an artificial channel, and also highlighted the complexity of their interactions. The artificial channel was effective in reducing the water retention time especially in areas close to the channel, while the drying climate trend has acted to increase the water retention time. The artificial channel enhanced the ocean intrusion, which had a mutual effect with the drying climate to increase the estuary salinity during the wet season, but it had opposite effect of reducing the hypersalinity during the dry season. The artificial channel increased the seawater fluxes and the salinity stratification, mostly in the Harvey Estuary, while the drying climate reduced the salinity stratification in the main body of the estuary. The changes in nutrient levels and habitat of pelagic and benthic communities related to hydrology are also discussed, which showed the communities are sensitive to the hydrologic changes. Consideration of the projected drying trend is essential in designing management plans associated with planning for environmental water provision and setting water quality loading targets.

**Data availability**

The datasets generated during the current study are available from the corresponding author on request.

**Author contribution**

All the authors contributed to the design of the study. Peisheng Huang carried the hydrology modelling work and prepared the
first draft of the manuscript. Karl Henig provided the catchment outputs of the inflow rates and nutrient concentrations. Jatin
Kala and Julia Andrys provided the WRF weather data. Matthew R. Hipsey was the project leader and provided technical and
financial supports. Jatin Kala and Matthew R. Hipsey helped to interpret the model data and write the article.

**Competing interests**

The authors declare no competing financial or non-financial interest.

**Acknowledgement**

The authors would like to thank three anonymous reviewers for their valuable and constructive comments. This research was
supported by the Australian Research Council Linkage Program (LP150100451).

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
