# Peer review of "of hydrological change in a large shallow lagoon"

_Hydrology and Earth System Sciences, 2020_

## Referee Comment (RC1) · Anonymous Referee #1 · 23 Jun 2020

**Review of "Climate change overtakes coastal engineering as the dominant driver of hydrological change in a large shallow lagoon" by Huang et al.**

In this study, the authors systematically investigated the responses of tidal hydrodynamics in terms of water retention time (including the water age and bulk flushing time), salinity and stratification to the drying climate trend (decrease in river flow inputs) and the opening of a large artificial channel. The obtained results are of particular importance for developing the corresponding sustainable water resources management strategies in such a large shallow lagoon system. However, there are still some major concerns that should be carefully addressed in order to improve the quality of this manuscript.

**Major concerns:**

1.  It should be noted that the Peel-Harvey lagoon (or estuary) is a typical tide-dominated system although it experiences a micro-tidal regime with tidal range generally less than 1 m. This is mainly due to the large estuarine surface area (133 km$^2$) and hence the large estuarine volume (187.5 GL). This indicates that the tidal hydrodynamics is generally featured by both the seasonal change and the spring-neap change. For the time being, the main results only focus on the seasonal change in tidal hydrodynamics, while the spring-neap change did not investigate at all.

2.  For the spring-neap change in the water retention time, it would be worth exploring the impacts of residual water currents on spatial-temporal variation in the water age and bulk flushing time. For instance, the difference of the residual water currents during the spring tide period before and after the opening of the artificial channel can be used to show the underlying mechanism of the change in water retention time. Similar results can be obtained for the neap tide period.

3.  It is noted that the morphological change during the study period 1970-2016 is neglected in the hydrodynamical model. It is better to clarify that such an assumption is reasonable.

4.  Meanwhile, it is worth noting that the water quality (such as salinity and stratification) in the Peel-Harvey estuary was dramatically impacted by the urban

development and the agricultural development in the upstream catchments. It appears that the authors also neglect these two factors in the hydrodynamical model. Some explanations can be provided in order to support the current results.

5. With regard to the riverine flow rate reduction, to what extend the river damming affects the river flow? As we know, both the Serpentine and Harvey Rivers are dammed in the upstream catchment.

**Some Minor concerns:**

1. In the title, 'hydrologic'→'hydrological'?

2. It is better to use the SI units for the whole paper. For instance, replacing GL with $m^3$ for the volume.

3. Figure 1: Add the north arrow and define the 'mAHD' in the main text.

4. Line 150: It is better to define the water age $\tau$ before mentioning it.

5. Lines 252-253: Why using 1990 and 1998 for a comparison? It is better to clarify the choice.

6. Figure 8: It is better to show some contour lines indicating the exact numbers.

7. Figure 9 and Figure 11: the color is not easy to distinguish.

8. Lines 471-472: It is better to define the TN and TP before using the abbreviations.

---

## Referee Comment (RC2) · Anonymous Referee #2 · 2 Jul 2020

**General comments**

In this manuscript, titled "Climate change overtakes coastal engineering as the dominant driver of hydrologic change in a large shallow lagoon", the authors describe the application of an unstructured modelling system to investigate hydrodynamics in the Peel-Harvey Estuary-Lagoon. Even if some may modelling studies dealing with lagoon's hydrodynamics have been already published, I particularly enjoyed reading this paper, which is clear, to the point and most interesting. The applied numerical model was properly applied and model results correctly presented and discussed. I particularly appreciate the multi-year investigation to separate the effects of climate change

and engineering interventions. I recommend publication, subject to the authors addressing the major comments made below.

**Specific concerns**

- Even the model has been validated, the authors did not carry out any calibration of the model parameters. The authors adopt bottom drag coefficient values based on the area type and the estimated biomass of aquatic vegetation within the cell. The selected values are probably retrieved from previous studies and not calibrated for the specific site. To my opinion every model application need a calibration phase were the most important model parameters are properly tuned (as also highlighted by the sensitivity tests ). Therefore, I suggest to perform a model calibration. As the

- I suggest including some general information about tide characteristics, average freshwater discharge and main wind regimes in PHE in the site description section (2.1).

- A detailed description of the open sea boundary conditions used in the simulations is needed.

- Please provide a more detailed description of the retention time computation (number of replicas per year, boundary conditions, initial conditions, treatment of the tail of the concentration decay when the simulation is shorted than the retention time, …). The work of Li et al. (2019) is not present in the reference list.

- Since the author is already computing the water retention time and the bulk flushing time, I strongly suggest to investigate the variation of the mixing efficiency of

the lagoon. This will allow the author to investigate the effect of climate change and cut-opening not only on the sea-lagoon exchange (flushing), but also on the internal mixing processes. As far as I understood, the retention time computed by the author is similar to the water renewal time estimated by Umgiesser et al. (2014). According to Umgiesser et al (2014), the ratio between the bulk flushing time and the mean renewal time can be interpreted as an index of the mixing behaviour of the basin (i.e. mixing efficiency, ME). ME ranges between 0 and 1 and is equal to 1 in case of a fully mixed system (renewal time becomes equal to flushing time). In the theoretical case of ME = 0, the water masses entering the lagoon do not mix at all with the inner waters, and the renewal time goes to infinity.

- In commenting the possible future changes in PHE hydrodynamics, please consider also that these coastal environments can act as sentinel systems for observation of global change (see ad example Ferrarin et al., 2014).

**Minor comments**

- Change hydrologic to hydrological.

- Line 13-15: I suggest to remove this statement since is not valid in general.

- For the water inflow rate and fluxes I would suggest to use m3 instead of GL.

- I suggest to remove Figure 11, because the results are clearly explained in the text.

**References**

Ferrarin, C., M. Bajo, D. Bellafiore, A. Cucco, F. De Pascalis, M. Ghezzo, and G. Umgiesser (2014a), Toward homogenization of Mediterranean lagoons and their loss of hydrodiversity, Geophys. Res. Lett., 41(16), 5935–5941, doi: 10.1002/2014GL060843.

Umgiesser, G., C. Ferrarin, A. Cucco, F. De Pascalis, D. Bellafiore, M. Ghezzo, and M. Bajo (2014), Comparative hydrodynamics of 10 Mediterranean lagoons by means of numerical modeling, J. Geophys. Res. Oceans, 119(4), 2212–2226, doi: 10.1002/2013JC009512.

---

## Referee Comment (RC3) · Anonymous Referee #3 · 8 Jul 2020

**Review of "Climate change overtakes coastal engineering as the dominant driver of hydrologic change in a large shallow lagoon " by Huang et al.**

In this manuscript, authors assessed the impacts of climate change and artificial channel on the water retention time, salinity and stratification of the Peel-Harvey Estuary (a large chocked-type lagoon) in 1970-2016, and their evolvement under current climate projections, based on a 3D finite-volume hydrodynamic model. There are some issues which are as follows.

**Major Points:**

1. In introduction section, authors mainly introduced the importance of the topic and what they did. However, the related work of other researchers was not introduced.

2. In the Methods section, it is better that all data used and their resources were introduced together. It seems that some data was introduced, but some not. In page 7, lines 202-203, "Gauged flow rate data for the Murray River, Serpentine River and Harvey River were applied to the hydrodynamic model  whenever they are available. For the missing periods in the gauged flows and the ungauged drains, the output from the Source ...". Which data is available, and which is not available? Which periods are the missing periods? Where are the ungauged drains?

3. About the meteorological inputs of the model, there are various data sources: station Halls Head before 1981, climate model simulations 1981-2000, and station Mandurah since 2001. The different sources of data could influence the results.

4. There is no calibration of the model parameters.

5. Topography is also an important factor affecting the hydrology of an estuary. Between 1970 and 2016, how did the topography of the lagoon and the natural channel change and affect the hydrology? There is no anything about this.

6. The results for 2040 and 2060 based on projected climate seem to be too simple. And there is no any explanation.

7. In 4.1 section, the contents in paragraphs 1, 2 and 3 mostly repeated the results. Here, the compare between your results and other related research should be shown.

**Minor points:**

1. All important stations or locations mentioned in the manuscript should be occurred in Figure 1.
2. GL and mAHD should be changed to international units.
3. Impacts on the stratification were shown in the results and conclusions. Why were they not mentioned in abstract?
4. The citation of reference is disorder. If several references are cited together, they should be put in order according to publishing year.
5. In page 6, line 186, WA region means Western Australia? The indication of this abbreviation is not seen.
6. In Figure 1 "Peel Estuary" is indicated. However, in the text and other figures only "Peel Inlet" can be seen. Are they the same location? If yes, in Figure 1, text, and other figures they should be the same.
7. Page 10, lines 277-279, authors said "the tide elevations in the ocean showed similar characteristics in 1990 and 1998 in terms of the annual mean sea level (-0.071 mAHD and -0.027 mAHD in 1990 and 1998, respectively) and tidal range (both < 1 m)".  The plot a of Figure 4 shows the detailed sea levels in 1990 and 1998. Why did you only compare the annual mean sea level? It can be seen from plot a that the sea levels in 1998 also had a wider range variation, similar to the estuary surface elevation.
8. Caption of Figure 4 is not proper. It looks like three figures.
9. Table 3: (1) In caption of the table, "Summary" should be deleted. (2) About the performance of salinity, it can be seen that errors after 1998 are clearly larger. Why?
10. Caption of Figure 10, "The darker symbols indicate the years with accidental summer rainfall  events and caused the catchment inflows higher than 15 GL".  In this sentence "and caused" seems syntax error.

---

## Editor Comment (EC1) · Hubert H.G. Savenije (Editor) · 8 Jul 2020

We have received three reviews which raise pertinent questions. I request the authors to provide answers to these reviews so as to provide an opportunity for further discussion.
* * *

---

## Short Comment (SC1) · 13 Jul 2020

Dear Hubert,

Thanks for the reminding of the response to the reviewers' comments. I have been proactively working on the response, though some of the comments required further data analysis that took some time. I have uploaded the response to the Reviewer #1. The response to other two reviewers are also drafted and will be sent out to the co-authors for feedback shortly.

Thanks, Best regards,

[Figure]

Peisheng Huang

---

## Author Comment (AC1) · 13 Jul 2020

**Response to the comments of reviewer #1:**

We thank the reviewer for the time examining the manuscript and the valuable comments that have provided insights on how to improve it. We have carefully addressed the comments in the following response and provided some further analysis to clarify the reviewer's major concerns. The comments of the reviewer are in black and our response are in blue.

**General comments:**

In this study, the authors systematically investigated the responses of tidal hydrodynamics in terms of water retention time (including the water age and bulk flushing time), salinity and stratification to the drying climate trend (decrease in river flow inputs) and the opening of a large artificial channel. The obtained results are of particular importance for developing the corresponding sustainable water resources management strategies in such a large shallow lagoon system. However, there are still some major concerns that should be carefully addressed in order to improve the quality of this manuscript.

**Major concerns:**

1.  It should be noted that the Peel-Harvey lagoon (or estuary) is a typical tide-dominated system although it experiences a micro-tidal regime with tidal range generally less than 1 m. This is mainly due to the large estuarine surface area (133 km$^2$) and hence the large estuarine volume (187.5 GL). This indicates that the tidal hydrodynamics is generally featured by both the seasonal change and the spring-neap change. For the time being, the main results only focus on the seasonal change in tidal hydrodynamics, while the spring-neap change did not investigate at all.

**Response:** We agree that the spring-neap tide change is an important driver of the changes in hydrology over the short-term scale. This study focused on examining the long-term changes in hydrology, considering the seasonal and inter-annual features of the estuary in response to climate change and catchment inputs, and therefore the changes in the system between the spring and neap times of the tidal cycle were not reported. However, our model does resolve these variability at this scale and have therefore

explored the tidal cycle impacts on the hydrology in more detail in the response to the following comment related to the spring-neap changes in water retention time.

2. For the spring-neap change in the water retention time, it would be worth exploring the impacts of residual water currents on spatial-temporal variation in the water age and bulk flushing time. For instance, the difference of the residual water currents during the spring tide period before and after the opening of the artificial channel can be used to show the underlying mechanism of the change in water retention time. Similar results can be obtained for the neap tide period.

**Response:** Noting that the system experiences a micro-tidal regime, we agree the spring-neap tide is a key driver of the short-term hydrological changes in the estuary, and therefore present a further investigation of this. We first analyzed the tide constituents of the Fremantle tide elevation record with the U-Tide utilities (Codiga, 2011). The results, as shown below in Table 1, identified the lunar diurnal constituents (K1, O1) contribute most of the tide potential energy, followed by the solar diurnal constituent (P1), principal lunar semidiurnal constituent (M2), and principal solar semi-diurnal constituent (S2).

**Table 1. Principal tidal constituents for Fremantle tide record**

| Constituents | Potential Energy (%) | Amplitude (m) | Greenwich phase lag (degrees) | Frequency (cycles per hour) |
|---|---|---|---|---|
| K1 | 61.08 | 0.156 | 324 | 0.0418 |
| O1 | 25.95 | 0.101 | 308 | 0.0387 |
| P1 | 6.06 | 0.049 | 314 | 0.0416 |
| M2 | 3.53 | 0.0374 | 323 | 0.0805 |
| S2 | 3.37 | 0.0365 | 334 | 0.0833 |

We then adopted the Eulerian time-mean method to calculate the residual current by averaging the currents during one or more tidal periods:

$$[\bar{u}, \ \bar{v}] = \frac{1}{T}\int_0^T [u, v]\,dt$$

where *u* and *v* represent the eastward and northward components of current; $\bar{u}$ and $\bar{v}$ are the Eulerian residual current; and T is the averaging period, here set to 24 hours given the dominant tidal period is close to 24 hours and the model output is in 2-hour interval. Eight time slots have been selected to investigate the residual currents and water retention time in the period of spring and neap tides (Figure 1). The selected time slots covered the time in a dry month (February) and a wet month (August) in years before and after the construction of the Dawesville Cut (1990 & 1998). Note that the Eulerian time-mean method we used here is relatively simple with primary aim to investigate the current circulation in selected times. Many analytical methods had been proposed to investigate the tide-induced Lagrangian and Eulerian mean circulations in coastal environments (e.g. Cheng, 1996; Pattiaratchi et al., 1996; Wei et al., 2004). But that requires different model settings to our modelling studies and is beyond the current research scope.

[Figure]

**Figure 1. Selected time slots for investigating the spring tide and neap tide on the water retention time. Each time slot spans 24 hours.**

The calculated daily residual currents over each time slot at the surface and bottom layers (Figure 2) suggested that the opening of the Dawesville Cut has a strong impact on the residual currents. In the year of 1990, the surface residual currents in the Harvey Estuary were mostly moving northward, and regional-scale residual circulation around the Peel Inlet can be observed. Whilst in the year of 1998, strong residual currents via the Dawesville Cut and occasional circulation around the northern Harvey Estuary are observed. The results also indicated the residual current speeds in the shallow water of the basins were relatively smaller than that in the deeper water.

[Figure]

**Figure 2. Plain views of daily residual currents in the selected time slots (as indicated in Figure 1). The total current speed is indicated with color scales from 0 – 0.1 m/s.**

We further analyzed the impacts of the spring tide and neap tide on the modelled water retention time by averaging the modelled water age in each of the selected time slots (Figure 3). The average water age presented a clear temporal difference between seasons,

and a spatial difference between the lagoons. The spatial distribution pattern of the water age is coincident with the residual current, e.g., areas around the channel mouth experienced more flushing, and the shallow water in the basins generally has higher water age than the deeper water.

[Figure]

**Figure 3. Plain views of average modelled water age in the selected time slots (as indicated in Figure 1).**

However, the difference of the water age between the spring and neap tide periods is relatively small when compared to the spatial and seasonal variations. A quantitative comparison of the water age between the spring tide and neap tide periods is shown in Figure 4, which shows the average water age in four areas of the estuary: east Peel Inlet, west Peel Inlet, north Harvey Estuary, and south Harvey Estuary. Note the water retention time is modelled using the 'water age' method, which is the time of a water parcel staying in the model domain since entering the domain from ocean and catchment input, so the impacts of tide excursion on the water age is compounded with the freshwater flushing. In this consideration, the summer time (February) with less freshwater input is more suitable to explore the tide impacts on the water age. In the year of 1990, the maximum difference in the water age in four areas is less than 5 days as observed in the Harvey

Estuary. Whilst in the year of 1998, the maximum difference of the water age between the spring tide and neap tide is about 10 days, as observed in north Harvey Estuary. These results indicate the spring-neap change can affect the hydrology over the short-term scale. However, the difference of water age between the spring tide and neap tide is still small when compared to the impacts of the opening of the Dawesville Cut (~20-110 days) and the reduced flow from 1970 to present (~50-100 days).

In summary, the residual current analysis has been shown to be a useful method to investigate the water transportation and circulation within the lagoons, and to explain the difference of water retention between the shallow and deep water. The results also suggested that the difference of modelled water age between the spring tide and neap tide periods were relatively small when compared to the impacts by the opening of the Dawesville Cut and reduced flow in the past decades. The above discussion on the spring-neap changes in the hydrology will be drafted and added into the manuscript in the revision.

[Figure]

EPI: East Peel Inlet; WPI: West Peel Inlet; NHE: North Harvey Estuary; SHE: South Harvey Estuary.

**Figure 4. Average water age during the spring tide and neap tide periods for selected time slots in four major regions of Peel-Harvey Estuary: east Peel Inlet; west Peel Inlet; north Harvey Estuary; south Harvey Estuary.**

3. It is noted that the morphological change during the study period 1970-2016 is neglected in the hydrodynamical model. It is better to clarify that such an assumption is reasonable.

**Response:** Thanks for pointing this out. We agree it is important to discuss the morphological changes in the study site as it is a large but shallow lagoon. In the modelling simulations, we assumed there the morphological change over the study period were not significant, except through the construction of the Dawesville Cut. The morphology data we used for the model was the latest morphology dataset from the Western Australia Department of Water, obtained in year 2016 (integrated DEM at 2m resolution).

The estuary morphology over the study period may have been modified by: (1) changes to the net sedimentation of particles; and (2) dredging activities related to marina and navigation channel developments. The net sedimentation rates in the Peel-Harvey Estuary had been investigated by a few early research. Gabrielson and Lukatelich (1985) estimated a net sedimentation rate of about 0.4-1.5 mm/year in the Peel Inlet and 2.9-6.7 mm/year in the Harvey Estuary; Hodgkin et al. (1980) estimated an overall rate of sediment deposition of 0.3 mm/year in the estuary. Assuming the rate is constant, the maximum total sediment deposition is about 75 mm in the Peel Inlet and 335 mm in the Harvey Estuary, over a period of 50 years. Note these rates were estimated in the time before the Dawesville Cut was constructed. After the year of 1994 when the Dawesville Cut was constructed, the sediment deposition in the system, especially in the Harvey lagoon, was expected to decrease due to higher tidal flushing. As we illustrated in the manuscript, a change in the tide elevation of ±0.15m, which could be theoretically equivalent to the change in the depth of the estuary, would have caused a small change to the hydrology compared to that introduced by the opening of the artificial channel and the reduced flow. Therefore we assumed the impact of sediment deposition on the morphology is small. The reduced flow rates over the course of the study period would also lead to a reduction in sediment loading.

The development of canal estates and navigation channels would have further changed the local morphology, but is expected to only slightly modify the estuary hydrology in

the regional scale. For example, The Yunderup navigation channel, located at the east side of the Peel Inlet, is one of the more significant dredging projects in the Peel-Harvey Estuary in the past decades. The Yunderup channel has a length of ~3km (mostly in the canal estate and shallow water areas) and a width of ~50m. The total area of this channel is ~0.015 km$^2$, which is negligible when compared to the area of the east Peel Inlet of 33.5 km$^2$. We therefore assume the changes brought about by the local dredging activities are negligible in analyzing the estuary hydrology when looking at the average properties over the regions.

We will integrate the above discussion into the manuscript to clarify the assumptions of using the estuary morphology data.

4. Meanwhile, it is worth noting that the water quality (such as salinity and stratification) in the Peel-Harvey estuary was dramatically impacted by the urban development and the agricultural development in the upstream catchments. It appears that the authors also neglect these two factors in the hydrodynamical model. Some explanations can be provided in order to support the current results.

**Response:** We agree that the catchment development, especially the urban development and the agriculture development will impact the water quality as well as the flows, though we note the urban expansion is mainly on the coastal edge which are minor sub-catchments to the system. We have briefly discussed this point in the manuscript (Section 4.3 – Uncertainty of future hydrology), but we will expand our discussion on the impacts of catchment development on the hydrology and water quality.

Different catchment developments had a combined effect on the flows. For example, the land clearing is expected to increase the streamflow, while local drainage changes have led to water diversions and reductions of the inflows. The Peel-Harvey catchment modelling report (Kelsey, 2011) estimated the land clearing to increase the annual streamflow by 290 GL, while water supply and irrigation dams to decrease flows by 145 GL, and the drain activity to divert about 100 GL, leading to a net increase of about 45GL. The net change in the streamflow is relatively small when compared to the reduced annual flows entering the Peel-Harvey Estuary from 1970 (846.4 GL) to 2016 (514.4 GL). In the

hydrologic models we have used the gauged flow data combined with the catchment model outputs, therefore the changes of the flow rates due to the catchment development have been accounted for in the model settings and our analysis. Therefore, we expect the reduced inflow was a key factor of reducing the nutrient loads from catchment, that subsequently reduced the nutrient concentrations in the estuary. We will further clarify this during our revision of the manuscript.

5. With regard to the riverine flow rate reduction, to what extend the river damming affects the river flow? As we know, both the Serpentine and Harvey Rivers are dammed in the upstream catchment.

**Response:** This is another critical point that need to be clarified of the catchment inflows and we sincerely thank the reviewer for pointing this out.

There are 15 dams in the Peel-Harvey catchment (Table 2). The total catchment area to be dammed is 1,283 $km^2$, which is about 12% of the total Peel-Harvey catchment (10,671 $km^2$) (Kelsey, 2011; Hennig et al., in prep). Most of the dams were completed in the time before 1970s when our study period started. The latest catchment modelling results (Hennig et al., in prep) showed that the average annual flow in the years of 2006-2015 from unrestricted catchments was 369 GL/yr while the flow from dammed catchments was 36 GL/yr, which would amount to an additional 10% increase in annual flow if these dams didn't exist. It is understood that no dams in the catchment have planned environmental water releases and it is expected that any water releases would either be a small proportion of flow. All dams except the North and South Dandalup dams have downstream flow measurement that was included in the estuary model. Thus, the effect of dam water releases would be included in the estuary model for the period where there is flow measurement. It is also understood that there are no plans to construct new dams in the catchment due to the considerable reduction in rainfall and streamflow in recent years. New water sources will likely come from unallocated groundwater (<10GL), desalination or wastewater reuse. We will add the above discussion into the manuscript and clarify the uncertainty caused by the dam activities.

**Table 2. Dams in the Peel-Harvey catchment. Re-printed from Table 2.4 of the catchment report of Kelsey (2011).**

| Dam | Completion year | Maximum capacity |
| --- | --- | --- |
| Serpentine | 1961 | 137667 |
| Serpentine Pipehead | 1957 | 2625 |
| North Dandalup | 1994 | 74849 |
| North Dandalup Pipehead | 1970 | |
| Conjurunup Pipehead | 1992 | 180 |
| South Dandalup | 1974 | 130000 |
| South Dandalup Pipehead | 1971 | |
| Waroona | 1966 | 15173 |
| Drakes Brook | 1931 | 2290 |
| Samson Brook | 1941 | 7993 |
| Samson Brook Pipehead | | |
| Logue Brook | 1963 | 24321 |
| Stirling | 1948 | 53769 |
| Stirling Pipehead | 1920 | |
| Harvey | 1916 | 56441 |

**Some Minor concerns:**

Thank you for these suggestions, we will update all these in the revision.

1. In the title, 'hydrologic'→'hydrological'?

**Response:** Comment accepted.

2. It is better to use the SI units for the whole paper. For instance, replacing GL with m3 for the volume.

**Response:** Comment accepted. We will check through and use the SI units consistently in the manuscript.

3. Figure 1: Add the north arrow and define the 'mAHD' in the main text.

**Response:** Comment accepted.

4. Line 150: It is better to define the water age τ before mentioning it.

**Response:** Comment accepted.

5. Lines 252-253: Why using 1990 and 1998 for a comparison? It is better to clarify the choice.

**Response:** The reason to use 1990 and 1998 for a comparison is that they are two closest years in all the selected modelling years to the year of the opening of the Dawesville Cut in 1994. They also had similar seasonal signals in the catchment inflows, and similar average tide elevation over the year. So we have selected these two years to illustrate the impacts of the Dawesville Cut without these other confounding factors, and to demonstrate the capability of the model to capture the changes in the hydrological features. We will add the explanation in the revision to clarify the use of the two years.

6. Figure 8: It is better to show some contour lines indicating the exact numbers.

**Response:** Comment accepted.

7. Figure 9 and Figure 11: the color is not easy to distinguish.

**Response:** Comment accepted. We will adopt a better color scheme for these figures.

8. Lines 471-472: It is better to define the TN and TP before using the abbreviations.

**Response:** Comment accepted.

**Reference:**

Codiga, D.L., 2011, Unified Tidal Analysis and Prediction Using the UTide Matlab Functions. Technical Report 2011-01. Graduate School of Oceanography, University of Rhode Island, Narragansett, RI. 59pp. ftp://www.po.gso.uri.edu/pub/downloads/codiga/pubs/2011Codiga-UTide-Report.pdf

Kelsey, P., Hall, J., Kretschmer, P., Quinton, B. & Shakya D., 2011, Hydrological and nutrient modelling of the Peel-Harvey catchment, Water Science Technical Series, Report no.33, Department of Water, Western Australia.

Gabrielson, J.O., & Lukatelich, R.J., 1985, Wind related resuspension of sediments in the Peel-Harvey estuarine system. Estuarine, Coastal Shelf Science 20, 135–145.

Hodgkin, E.P., Birch, P.B., Black, R.E., and Humphries R.B., 1980, The Peel-Harvey estuarine system study, Report no. 9, Department of Conservation and

Environment, Western Australia.

Pattiaratchi, C., Bowman, M. J. (Ed.), & Mooers, C. N. K. (Ed.), 1996. *Mixing in Estuaries and Coastal Seas*. Washington, D.C.: American Geophysical Union.

Wei, H., Hainbucher, D., Pohlmann, T., Feng, S.Z., Suendermann J., 2004, Tidal-induced Lagrangian and Eulerian mean circulation in the Bohai Sea, Journal of Marine Systems, 44 (2004), pp. 141-151.

Cheng, R., 1990, Residual currents and long-term transport. Springer, Berlin, 544 p.

Hennig, K., Kelsey, P., Hall, J., Gunaratne, G.G., Robb, M., in preparation, Hydrological and nutrient modelling of the Peel-Harvey estuary catchment (2006–15), Water Science Technical Series, Aquatic Science Branch, Department of Water and Environmental Regulation, Perth, Western Australia.

---

## Author Comment (AC2) · 22 Jul 2020

**Response to the comments of reviewer #2:**

We thank the reviewer for the time examining the manuscript and the valuable comments that have helped to improve the manuscript. We have carefully addressed the comments in the following response and provided some further analysis to clarify the reviewer's major concerns. The comments of the reviewer are in black and our response are in blue.

**General comments:**

In this manuscript, titled "Climate change overtakes coastal engineering as the dominant driver of hydrologic change in a large shallow lagoon", the authors describe the application of an unstructured modelling system to investigate hydrodynamics in the Peel-Harvey Estuary-Lagoon. Even if some may modelling studies dealing with lagoon's hydrodynamics have been already published, I particularly enjoyed reading this paper, which is clear, to the point and most interesting. The applied numerical model was properly applied and model results correctly presented and discussed. I particularly appreciate the multi-year investigation to separate the effects of climate change and engineering interventions. I recommend publication, subject to the authors addressing the major comments made below.

**Specific concerns:**

 Even the model has been validated, the authors did not carry out any calibration of the model parameters. The authors adopt bottom drag coefficient values based on the area type and the estimated biomass of aquatic vegetation within the cell. The selected values are probably retrieved from previous studies and not calibrated for the specific site. To my opinion every model application need a calibration phase were the most important model parameters are properly tuned (as also highlighted by the sensitivity tests). Therefore, I suggest to perform a model calibration.

**Response:** Thank you for highlighting this aspect of the model setup. Ideally, we could adopt an automated calibration approach, aiming to minimize error in key model predictions using an objective function and a pre-determined acceptable criteria for model acceptability (e.g. Doherty and Johnston, 2003; Arhonditsis et al., 2008; Bahremand and Smedt, 2010). This approach has yet to receive widespread up-take in the hydrodynamic

modelling community, particularly where 3-D models are employed to resolve variability in stratification, due to considerable computational burden running these models hundreds or thousands of times. Given the complex nature of the model domain that we wanted to adopt to resolve the river reaches to the tidal limit, and an individual run-time exceeding one-day per year, we therefore could not adopt an automatic optimization approach.

Instead, we adopted a structured hierarchical approach to calibration, similar to those described in Muleta and Nicklow (2004) and Hipsey et al. (2020), to manually calibrate the model. We first identified the key parameters of importance to the hydrology in the current study based on literature review and prior expert knowledge. In this stage, the key parameters were identified to be the bottom drag coefficient (which can vary spatially), the light extinction coefficient, bulk aerodynamic coefficients, and the mixing scheme options associated with the vertical turbulence model (in this case this is parameterized through the GOTM plugin). In the second stage, we evaluated a matrix of simulations, each with pre-determined parameter vectors and model options, by assessing model performance of each simulation against the observed salinity and temperature data at six stations within the estuary (at both surface and bottom levels), and the water elevation at the center of the Peel Inlet. For these we tabulated a summary of error metrics R, NSE and PBIAS, and used this to identify the final parameter options used for the validation simulations that were presented in the paper. The assessment was targeted, and included comparing performance of mixing models, against metrics relevant to the analysis such as stratification strength and hyper-salinity associated with evapo-concentration We also acknowledge issues in boundary condition data may affect the calibration, and we therefore spent considerable effort on the data quality control of the time-series of tide, weather, and catchment inputs to the model. In addition, the sensitivity of predictions to the selected environmental factors of air temperature, sea level mean height, and the bottom drag coefficient were performed.

We acknowledge that this approach is not necessarily providing the most optimum parameter set from a mathematical point of view, however, given other uncertainties in the spatial maps of vegetation (and therefore benthic drag) and potential error or bias in some of the assumed boundary conditions, it is our view that the model performance is close to the optimum and sufficiently accurate for the scale of our assessment. To address this in the paper, we therefore propose to add in the revised version a brief summary of the calibration approach and the results as supplemental material to this manuscript, and provide an improved discussion that describes to the known uncertainties and limitations of the model in this regard.

I suggest including some general information about tide characteristics, average freshwater discharge and main wind regimes in PHE in the site description section (2.1).

**Response:** Comment accepted. We are rewording the paragraph in the site description section to include more detailed description as below:

"The estuary experiences a micro-tide regime, with a range < 1m. The tide is dominated by the lunar diurnal constituents (K1, O1) contributing 87% of the tide potential energy, followed by the solar diurnal constituent (P1), principal lunar semidiurnal constituent (M2), and principal solar semi-diurnal constituent (S2) (Table 1). The coastal catchment of the estuary is drained by three major river systems: the Serpentine, Murray and Harvey Rivers (on average contributing 16.4%, 46.5%, and 30.8% to the total flow, respectively), and numerous minor drains (contributing 6.3% to the total flow) (Kelsey et al., 2011). Gauged flow rate data for Murray River were available from 1970 to present, while for Serpentine River and Harvey River were available from 1982 to present. For the missing periods in the gauged flows (year 1970 and 1978 for Serpentine River and Harvey River) and the ungauged drains, the output from the Source (eWater®) catchment modelling platform (Kelsey et al., 2011; Welsh et al., 2013), operated by the Western Australia 204 Department of Water and Environmental Regulation, was used."

| Constituents | Potential  | Amplitude | Greenwich | Frequency (cycles |
|--------------|------------|-----------|-----------|-------------------|
|              | Energy (%) | (m)       | phase lag | per hour)         |
|              |            |           | (degrees) |                   |
| K1           | 61.08      | 0.156     | 324       | 0.0418            |
| 01           | 25.95      | 0.101     | 308       | 0.0387            |
| P1           | 6.06       | 0.049     | 314       | 0.0416            |

Table 1. Principal tidal constituents for Fremantle tide record

| M2        | 3.53 | 0.0374 | 323 | 0.0805 |
|-----------|------|--------|-----|--------|
| S2 | 3.37 | 0.0365 | 334 | 0.0833 |

The wind regime will be further explained in the data section. We used a combination of weather station records and regional weather models to produce the wind conditions, and we have undertaken a further investigation into the wind regimes from these three sources. As shown in the figure below, the wind regimes of these data sources showed similar distribution in wind magnitudes and directions, though the winds in the Mandurah station record are relatively smaller when compared to other two sources. We will integrate the figure in the revision and indicate the selection of meteorological sources could have influence the results in the years after 2001.

**Figure 1.** Rose plot of wind condition in years of (a) 1970-1980, obtained from the Halls Head weather station; (b) 1981-2000, obtained from the WRF weather model; and (c) 2001-2016, obtained from the Mandurah weather station.

3. A detailed description of the open sea boundary conditions used in the simulations is needed.

**Response:** Comment accepted. We will add an open sea boundary condition description in the model set up section as below: "The water level at the ocean boundary was specified

with the Fremantle gauge station record, while the velocity was calculated internally based on a radiation condition".

4. Please provide a more detailed description of the retention time computation (number of replicas per year, boundary conditions, initial conditions, treatment of the tail of the concentration decay when the simulation is shorted than the retention time, . . .). The work of Li et al. (2019) is not present in the reference list.

**Response:** Comment accepted. For each selected year, the modelling simulation started from 1st September of the previous year, giving a 4-month spin-up period, and the results from 1st January to the end of the selected year were used for analysis. The initial condition of water temperature and salinity was interpolated from the field data when they were available (years 1985-2016), except the years 1970 and 1978 when no field data was available and so the same initial condition as for 1985 was adopted.

Our study adopted a 'water age' method to calculate the water retention time, which is defined as the time of a water parcel has spent since entering the model domain through one of the boundaries (Zimmerman, 1988; Monsen et al., 2002). The modelled water age was initially set to be 0 across the domain, then the water age in each computational cell was computed as a conservative tracer subject to a constant increase with time (i.e.  $d\tau/dt = 1$ , where  $\tau$  is the water age, and t is the time, in addition to advection and mixing), with the boundary values of  $\tau$  set to 0. While the water age method has an advantage of presenting the spatial heterogeneity of water retention (Monsen et al., 2002), this method is different to the water renewal time, which is calculated by the rate of tracer decaying within the study domain. Therefore the treatment of the tail of the concentration decay is not applicable in our study (further explanation of the difference between the water age and water flushing time methods is provided in the response to the next comment related to mixing efficiency). We will seek to integrate this information more clearly into the model set up section in the revision.

Apologizes for the missing reference of Li et al., (2019). The work of Li et al. (2019) was cited in the methodology section as a reference for the water age method. The full description of this reference is "Li, Y., Feng, H., Zhang, H., Sun, J., Yuan, D., Guo, L.,

Nie, J., Du, J. (2019), Hydrodynamics and water circulation in the New York/New Jersey Harbor: A study from the perspective of water age, Journal of Marine Systems, 199, doi.org/10.1016/j.jmarsys.2019.103219". We will add this reference in the revision of the manuscript.

5. Since the author is already computing the water retention time and the bulk flushing time, I strongly suggest to investigate the variation of the mixing efficiency of the lagoon. This will allow the author to investigate the effect of climate change and cut-opening not only on the sea-lagoon exchange (flushing), but also on the internal mixing processes. As far as I understood, the retention time computed by the author is similar to the water renewal time estimated by Umgiesser et al. (2014). According to Umgiesser et al (2014), the ratio between the bulk flushing time and the mean renewal time can be interpreted as an index of the mixing behaviour of the basin (i.e. mixing efficiency, ME). ME ranges between 0 and 1 and is equal to 1 in case of a fully mixed system (renewal time becomes equal to flushing time). In the theoretical case of ME = 0, the water masses entering the lagoon do not mix at all with the inner waters, and the renewal time goes to infinity.

Response: Thank you for the recommendation of the reference of Umgiesser et al. (2014), which compared the water retention time in 10 Mediterranean lagoons. We have carefully read this reference and found it a valuable reference to this research. The work of Umgiesser et al. (2014) adopted two hydrodynamic time parameters, one is water renewal time (WRT) and the other water flushing time (WFT). In their models, they set the initial tracer concentration to be 1 across the study domain, then calculate the WRT and WRF as the rate of tracer reduction. Because these two time parameters use the same mathematical method to calculate the retention time, they can then define the mixing efficiency (ME) as the ratio between WFT and WRT, and compared this mixing character between different lagoons.

We have found the method in this study attractive to study the mixing in lagoon estuaries, though we note that the water age method we used is theoretically different to the water renewal time method, and they cannot be compared directly to the bulk flushing time (Zimmerman, 1988; Monsen et al., 2002). Although our current simulation outputs cannot reveal the mixing efficiency with the water age method, we find this concept is valuable to compare the mixing character before and after the construction of the artificial channel, as well as under wet and dry conditions. Therefore, we propose to carry further simulations using the tracer decay method so we can compute the WRT and WFT metrics in the Peel-Harvey Estuary. The inter-comparison of mixing efficiency changes affected by the climate and artificial channel will be presented and discussed, and placed in context of the Umgiesser et al. (2014) results.

6. In commenting the possible future changes in PHE hydrodynamics, please consider also that these coastal environments can act as sentinel systems for observation of global change (see ad example Ferrarin et al., 2014).

Response: Comment accepted. We thank the reviewer for the recommendation of the reference of Ferrarin et al., (2014). This paper is a valuable reference to the current manuscript. It showed that Mediterranean lagoons are sensitive to the climate change as they amplify the salinity and temperature changes expected for the open sea. It also showed that the coastal lagoon systems are under stress of not only the human activities but also the climate change. Therefore, the research of climate change on coastal systems has implications on more general scope. The Peel-Harvey Estuary, in this scope, can work as a valuable example for assessing climate change because it has a long historical record of its hydrology since 1980s, and the local system has experienced a notable change due to the consistent drying climate. We will integrate this response into the revision to enhance the discussion of the roles of lagoons in global climate change studies.

**Minor comments:**

1. Change hydrologic to hydrological

Response: Comment accepted.

**2. Line 13-15: I suggest to remove this statement since is not valid in general.**

**Response:** Thank you for pointing out this. We did not mean that artificially channel is a common engineering intervention of many estuarine lagoons. We will reword the text as: "The introduction of artificial channel is a fundamental engineering measure to enhance flushing and alter the hydrology of estuarine lagoon systems, however, the effects from the interaction of climate change with artificial channels have not been well evaluated."

**3. For the water inflow rate and fluxes I would suggest to use m3 instead of GL.**

**Response:** Comment accepted. We will check through the manuscript to use the SI units consistently.

**4. I suggest to remove Figure 11, because the results are clearly explained in the text.**

**Response:** Thank you for the suggestion. While we agree that the results of salinity stratification are explained in the text, we still think the Figure 11 can work to provide a quantitative evaluation of the salinity stratification in space and time, therefore help the readers to better understand the heterogeneity and evolution of stratification. As stratification is also a key character of the estuary hydrology, we would suggest to keep this figure in the manuscript, but subject to finalization of the discussion.

**Reference:**

- Arhonditsis, G. B., Perhar, G., Zhang, W., Massos, E., Shi, M., Das, A., 2008, Addressing equifinality and uncertainty in eutrophication models, Water Resour. Res., 44, W01420, doi:10.1029/2007WR005862.
- Bahremand, A., Smedt, F.D. 2010, Predictive Analysis and Simulation Uncertainty of a Distributed Hydrological Model, Water Resources Management volume 24, 2869– 2880.
- Doherty, J., Johnston, J.M., 2003. Methodologies for Calibration and Predictive Analysis of a Watershed Model, Journal of the American Water Resources Association (JAWRA), 39(2):251-265.
- Ferrarin, C., Bajo, M., Bellafiore, D., Cucco, A., De Pascalis, F., Ghezzo, M., Umgiesser G., 2014, Toward homogenization of Mediterranean lagoons and their loss of hydrodiversity, Geophys. Res. Lett., 41, 5935–5941, doi:10.1002/2014GL060843.

- Hipsey, M.R., Gal, G., Arhonditsis, G.B., Carey, C.C., Elliott, J.A., Frassl, M.A., Janse, J.H., de Mora, L., Robson, B.J. A system of metrics for the assessment and improvement of aquatic ecosystem models. Environ. Model. Soft 2020, 128, 104697.
- Kelsey, P., Hall, J., Kretschmer, P., Quiton, B. and Shakya, D.: Hydrological and nutrient modelling of the Peel-Harvey catchment, Water Science Technical Series, Report no. 33, Department of Water, Western Australia., 2011.
- Li, Y., Feng, H., Zhang, H., Sun, J., Yuan, D., Guo, L., Nie, J., Du, J., 2019, Hydrodynamics and water circulation in the New York/New Jersey Harbor: A study from the perspective of water age, Journal of Marine Systems, 199, doi.org/10.1016/j.jmarsys.2019.103219.
- Monsen, N. E., Cloern, J. E., Lucas, L.V., Monismith, S.G., 2002. A comment on the use of flushing time, residence time, and age as transport time scales. Limnology & Oceanography, 47:1545–1553.
- Muleta, M.K., Nicklow, J.W., 2004, Sensitivity and uncertainty analysis coupled with automatic calibration for a distributed watershed model, J. Hydrol., 306, 127–145.
- Umgiesser, G., Ferrarin, C., Cucco, A., De Pascalis, F., Bellafiore, D., Ghezzo, M., Bajo, M., 2014, Comparative hydrodynamics of 10 Mediterranean lagoons by means of numerical modeling, J. Geophys. Res. Oceans, 119, 2212–2226, doi:10.1002/2013JC009512.
- Welsh, W.D., Vaze, J., Dutta, D., Rassam, D., Rahman, J.M., Jolly, I.D., Wallbrink, P., Podger, G.M., Bethune, M., Hardy, M.J., Teng, J., Lerat, J., 2013, An integrated modelling framework for regulated river systems, Environ. Model. Softw., 39, 81– 102, doi:10.1016/j.envsoft.2012.02.022.
- Zimmerman, J.T.F., 1988. Estuarine residence times, p. 75–84. In B. Kjerfve [ed.], Hydrodynamics of estuaries. V. 1. CRC Press.

---

## Author Comment (AC3) · 24 Jul 2020

We thank the reviewer for the valuable comments and the time examining the manuscript. We have carefully addressed the comments in the following response and provided some further analysis and statements to clarify the reviewer's major concerns.

Please also note the supplement to this comment:
https://hess.copernicus.org/preprints/hess-2020-164/hess-2020-164-AC3-supplement.pdf

164, 2020.

---

## Author Response (AR2)

*07th October 2020*

Dear Hydrology and Earth System Sciences Editor,

**Re. Revision of hess-2020-164: Climate change overtakes coastal engineering as the dominant driver of hydrological change in a large shallow lagoon**

We thank you for the acceptance of our paper to be published in HESS. Here we upload a fully revised paper that includes all the corrections that you suggested in your revision comments, on the top of other changes to address all the comments from the reviewers. We have also went through the manuscript again to clean up the format of the reference list according to the HESS standard. The previous response to the reviewers' comments during the revision phase and a tracked-changes version are also enclosed in this document.

We thank you again for the significant time and effort that have gone into the discussion and helped to improve the paper.

Kind Regards

Peisheng Huang, on behalf of all co-authors.

*20th September 2020*

Dear Hydrology and Earth System Sciences Editor & Reviewers,

**Re. Revision of hess-2020-164: Climate change overtakes coastal engineering as the dominant driver of hydrological change in a large shallow lagoon**

We thank you for the opportunity for us to revise the paper, and are very grateful for the comments received during the discussion phase of the paper. We have found the comments very detailed and insightful and they have guided us to significantly improve this version of the manuscript.

We have already provided specific replies back to each of the reviewer comments in the discussion forum explaining our approach to the revision. A summary of the main issues identified were:

1. Potential impacts from the morphological change and catchment development to the results;
2. The future climate projection is simplistic;
3. Lack of details of the model calibration and validation;
4. Water transportation and mixing within the lagoon can be improved with the mixing efficiency method;
5. Issues with figure readability and captions;
6. Numerous typographical and minor editorial issues in the notation and units, plus some errors with reference citations and the formatting consistency in the reference list;

Here we upload a fully revised paper that includes significant changes to address all the comments, where possible. The major changes relate to: (1) a re-worked introduction to better introduce the related research advances; (2) a new discussion section to address the potential impacts from the morphological change and catchment development to our results; (3) a new discussion section to further discuss the future climate projection approach; (4) applications and discussion of water flushing time, water renewal time, and mixing efficiency to investigate the water transportation and mixing within the lagoon in a few theoretical scenarios; and (5) a supplementary document to provide the details of model calibration, validation, and also the model sensitivity to selected environmental drivers. Many of the issues relevant to issues 5-6 were resolved and the text has been cleaned up in this upload. Please refer to the individual responses for specific details. A tracked-changes version is at the back of this document.

We thank you again for the significant time and effort that have gone into the discussion and look forward to your decision.

Kind Regards

Peisheng Huang, on behalf of all co-authors.

**Response to the comments of reviewer #1:**

We thank the reviewer for the time examining the manuscript and the valuable comments that have provided insights on how to improve it. We have carefully addressed the comments in the following response and provided some further analysis to clarify the reviewer's major concerns, and indicate the relative revision in the revised manuscript. The comments of the reviewer are in black and our response are in blue.

**General comments:**

In this study, the authors systematically investigated the responses of tidal hydrodynamics in terms of water retention time (including the water age and bulk flushing time), salinity and stratification to the drying climate trend (decrease in river flow inputs) and the opening of a large artificial channel. The obtained results are of particular importance for developing the corresponding sustainable water resources management strategies in such a large shallow lagoon system. However, there are still some major concerns that should be carefully addressed in order to improve the quality of this manuscript.

**Major concerns:**

1. It should be noted that the Peel-Harvey lagoon (or estuary) is a typical tide- dominated system although it experiences a micro-tidal regime with tidal range generally less than 1 m. This is mainly due to the large estuarine surface area (133 km$^2$) and hence the large estuarine volume (187.5 GL). This indicates that the tidal hydrodynamics is generally featured by both the seasonal change and the spring- neap change. For the time being, the main results only focus on the seasonal change in tidal hydrodynamics, while the spring-neap change did not investigate at all.

**Response:** We agree that the spring-neap tide change is an important driver of the changes in hydrology over the short-term scale. This study focused on examining the long-term changes in hydrology, considering the seasonal and inter-annual features of the estuary in response to climate change and catchment inputs, and therefore the changes in the system between the spring and neap tidal cycles were not reported. However, our model does resolve these variability at this scale and have therefore explored the tidal cycle impacts on the hydrology in more detail in the response to the following comment related to the spring-neap changes in water retention time.

2. For the spring-neap change in the water retention time, it would be worth exploring the impacts of residual water currents on spatial-temporal variation in the water age and bulk flushing time. For instance, the difference of the residual water currents during the spring tide period before and after the opening of the artificial channel can be used to show the underlying mechanism of the change in water retention time. Similar results can be obtained for the neap tide period.

**Response:** We had explored the changes in the water age during the spring tide and neap tide in our previous response to this comments in the discussion forum during review phase (the details are also included below). In summary, the residual current analysis has been shown to be a useful method to investigate the water transportation and circulation within the lagoons, and to explain the difference of water retention between the shallow and deep water. However, the results also suggested that the difference of modelled water age between the spring tide and neap tide periods were relatively small when compared to the impacts by the opening of the Dawesville Cut and reduced flow in the past decades. We therefore did not include this discussion of the spring-neap tides into the revision. Instead, we used the residual current concept to compare the current characteristics between the Cut open/close and catchment inflow scenarios (in section 4.1 of the revised manuscript).

**Previous analysis results and response to the comment in the discussion forum during review phase:**

Noting that the system experiences a micro-tidal regime, we agree the spring-neap tide is a key driver of the short-term hydrological changes in the estuary, and therefore present a further investigation of this. We first analyzed the tide constituents of the Fremantle tide elevation record with the U-Tide utilities (Codiga, 2011). The results, as shown below in Table 1, identified the lunar diurnal constituents (K1, O1) contribute most of the tide potential energy, followed by the solar diurnal constituent (P1), principal lunar semidiurnal constituent (M2), and principal solar semi-diurnal constituent (S2).

**Table 1.** Principal tidal constituents for Fremantle tide record

| Constituents | Potential Energy (%) | Amplitude (m) | Greenwich phase lag (degrees) | Frequency (cycles per hour) |
|---|---|---|---|---|
| **K1** | 61.08 | 0.156 | 324 | 0.0418 |
| **O1** | 25.95 | 0.101 | 308 | 0.0387 |
| **P1** | 6.06 | 0.049 | 314 | 0.0416 |
| **M2** | 3.53 | 0.0374 | 323 | 0.0805 |
| **S2** | 3.37 | 0.0365 | 334 | 0.0833 |

We then adopted the Eulerian time-mean method to calculate the residual current by averaging the currents during one or more tidal periods:

$$[\bar{u}, \ \bar{v}] = \frac{1}{T} \int_0^T [u, v] dt$$

where $u$ and $v$ represent the eastward and northward components of current; $\bar{u}$ and $\bar{v}$ are the Eulerian residual current; and T is the averaging period, here set to 24 hours given the dominant tidal period is close to 24 hours and the model output is in 2-hour interval. Eight time slots have been selected to investigate the residual currents and water retention time in the period of spring and neap tides (Figure S1). The selected time slots covered the time in a dry month (February) and a wet month (August) in years before and after the construction of the Dawesville Cut (1990 & 1998). Note that the Eulerian time-mean method we used here is relatively simple with primary aim to investigate the current circulation in selected times. Many analytical methods had been proposed to investigate the tide-induced Lagrangian and Eulerian mean circulations in coastal environments (e.g. Cheng, 1996; Pattiaratchi et al., 1996; Wei et al., 2004). But that requires different model settings to our modelling studies and is beyond the current research scope.

[Figure]

**Figure S1.** Selected time slots for investigating the spring tide and neap tide on the water retention time. Each time slot spans 24 hours.

The calculated daily residual currents over each time slot at the surface and bottom layers (Figure S2) suggested that the opening of the Dawesville Cut has a strong impact on the residual currents. In the year of 1990, the surface residual currents in the Harvey Estuary were mostly moving northward, and regional-scale residual circulation around the Peel Inlet can be observed. Whilst in the year of 1998, strong residual currents via the Dawesville Cut and occasional circulation around the northern Harvey Estuary are observed. The results also indicated the residual current speeds in the shallow water of the basins were relatively smaller than that in the deeper water.

[Figure]

**Figure S2.** Plain views of daily residual currents in the selected time slots (as indicated in Figure S1). The total current speed is indicated with colour scales from 0-0.1 m/s.

We further analysed the impacts of the spring tide and neap tide on the modelled water retention time by averaging the modelled water age in each of the selected time slots (Figure S3). The average water age presented a clear temporal difference between seasons, and a spatial difference between the lagoons. The spatial distribution pattern of the water age is coincident with the residual current, e.g., areas around the channel mouth experienced more flushing, and the shallow water in the basins generally has higher water age than the deeper water.

[Figure]

**Figure S3.** Plain views of average modelled water age in the selected time slots (as indicated in Figure S1).

However, the difference of the water age between the spring and neap tide periods is relatively small when compared to the spatial and seasonal variations. A quantitative comparison of the water age between the spring tide and neap tide periods is shown in Figure S4, which shows the average water age in four areas of the estuary: east Peel Inlet, west Peel Inlet, north Harvey Estuary, and south Harvey Estuary. Note the water retention time is modelled using the 'water age' method, which is the time of a water parcel staying in the model domain since entering the domain from ocean and catchment input, so the impacts of tide excursion on the water age is compounded with the freshwater flushing. In this consideration, the summer time (February) with less freshwater input is more suitable to explore the tide impacts on the water age. In the year of 1990, the maximum difference in the water age in four areas is less than 5 days as observed in the Harvey Estuary. Whilst in the year of 1998, the maximum difference of the water age between the spring tide and neap tide is about 10 days, as observed in north Harvey Estuary. These results indicate the spring-neap change can affect the hydrology over the short-term scale. However, the difference of water age between the spring tide and neap tide is still small when compared to the impacts of the opening of the Dawesville Cut (~20-110 days) and the reduced flow from 1970 to present (~50-100 days).

[Figure]

EPI: East Peel Inlet; WPI: West Peel Inlet; NHE: North Harvey Estuary; SHE: South Harvey Estuary.

**Figure S4.** Average water age during the spring tide and neap tide periods for selected time slots in four major regions of Peel-Harvey Estuary: east Peel Inlet; west Peel Inlet; north Harvey Estuary; south Harvey Estuary.

3.  It is noted that the morphological change during the study period 1970-2016 is neglected in the hydrodynamical model. It is better to clarify that such an assumption is reasonable.

**Response:** Thanks for pointing this out. We agree it is important to discuss the morphological changes in the study site as it is a large but shallow lagoon. We had provided a literature review and discussion on this issue in our early response in the discussion forum. We have now included a new section in the discussion (section 4.3 - "Potential impacts from the morphological change and catchment development on the Peel-Harvey hydrology") in the revised manuscript to address the potential impacts from the morphological changes on the results.

4.  Meanwhile, it is worth noting that the water quality (such as salinity and stratification) in the Peel-Harvey estuary was dramatically impacted by the urban development and the agricultural development in the upstream catchments. It appears that the authors also neglect these two factors in the hydrodynamical model. Some explanations can be provided in order to support the current results.

**Response:** We agree that the catchment development, especially the urban development and the agriculture development will impact the water quality as well as the flows, though we note the urban expansion is mainly on the coastal edge which are minor sub-catchments to the system. We had briefly discussed this point in our previous response on the discussion forum, and now included a more detailed discussion in the section 4.3 ("Potential impacts from the morphological change and catchment development on the Peel-Harvey hydrology") of the revised manuscript to address this issue.

5.  With regard to the riverine flow rate reduction, to what extend the river damming affects the river flow? As we know, both the Serpentine and Harvey Rivers are dammed in the upstream catchment.

**Response:** This is another critical point that need to be clarified of the catchment inflows and we sincerely thank the reviewer for pointing this out. Please see the detailed response below. We have included the major points of this response in the section 4.3 ("Potential impacts from the morphological change and catchment development on the Peel-Harvey hydrology") of the revised manuscript.

There are 15 dams in the Peel-Harvey catchment (Table S2). The total catchment area to be dammed is 1,283 km$^2$, which is about 12% of the total Peel-Harvey catchment (10,671 km$^2$) (Kelsey, 2011; Hennig et al., in prep). Most of the dams were completed in the time before 1970s when our study period started. The latest catchment modelling results (Hennig et al., in prep) showed that the average annual flow in the years of 2006-2015 from unrestricted catchments was 369 GL/yr while the flow from dammed catchments was 36 GL/yr, which would amount to an additional 10% increase in annual flow if these dams didn't exist. It is understood that no dams in the catchment have planned environmental water releases and it is expected that any water releases would either be a small proportion of flow. All dams except the North and South Dandalup dams have downstream flow measurement that was included in the estuary model. Thus, the effect of dam water releases would be included in the estuary model for the period where there is flow measurement. It is also understood that there are no plans to construct new dams in the catchment due to the considerable reduction in rainfall and streamflow in recent years. New water sources will likely come from unallocated groundwater (<10GL), desalination or wastewater reuse.

**Table S2.** Dams in the Peel-Harvey catchment. Re-printed from Table 2.4 of the catchment report of Kelsey (2011).

| Dam | Completion year | Maximum capacity |
|---|---|---|
| Serpentine | 1961 | 137667 |
| Serpentine Pipehead | 1957 | 2625 |
| North Dandalup | 1994 | 74849 |
| North Dandalup Pipehead | 1970 | |
| Conjurunup Pipehead | 1992 | 180 |
| South Dandalup | 1974 | 130000 |
| South Dandalup Pipehead | 1971 | |
| Waroona | 1966 | 15173 |
| Drakes Brook | 1931 | 2290 |
| Samson Brook | 1941 | 7993 |
| Samson Brook Pipehead | | |
| Logue Brook | 1963 | 24321 |
| Stirling | 1948 | 53769 |
| Stirling Pipehead | 1920 | |
| Harvey | 1916 | 56441 |

**Some Minor concerns:**

1. In the title, 'hydrologic'→'hydrological'?

**Response:** Comment accepted.

2. It is better to use the SI units for the whole paper. For instance, replacing GL with m3 for the volume.

**Response:** Comment accepted. We have checked through the manuscript and use the SI units consistently in the revision.

3. Figure 1: Add the north arrow and define the 'mAHD' in the main text.

**Response:** Comment accepted.

4. Line 150: It is better to define the water age τ before mentioning it.

**Response:** Comment accepted.

5. Lines 252-253: Why using 1990 and 1998 for a comparison? It is better to clarify the choice.

**Response:** Comment accepted. These two years were compared because they have similar annual precipitation and catchment inflow rates, and tidal forcing characteristics in terms of the annual mean sea level and tidal range. Therefore, the comparison provided a valuable insight into the impacts of the artificial channel on the estuary environment. We have clarify the reasons of using these two years for comparison and the details of tidal characteristics in the section 3.1 ("Estuary response to the change in ocean connectivity") in the revised manuscript.

6. Figure 8: It is better to show some contour lines indicating the exact numbers.

**Response:** Comment accepted. We have now used contour colors and lines to indicate the numbers (Figure 9 in the revised manuscript).

7. Figure 9 and Figure 11: the color is not easy to distinguish.

**Response:** Comment accepted. We have improved the color scheme for these figures (Figure 10 and 12 in the revised manuscript).

8. Lines 471-472: It is better to define the TN and TP before using the abbreviations.

**Response:** Comment accepted.

**Response:** We agree that meteorological inputs from different sources would influence the results due to site specific biases they may have. We have now added a rose plot (same as Figure S5 below) and description of the wind regimes from various data source in the revised manuscript (section 2.2.4 – "Climate change context and simulation rationale"), and added the locations of weather stations in Figure 1. Though various sources of climate data were used, the wind regimes of these data sources showed similar distribution in wind magnitudes and directions. The winds in the Mandurah station record are relatively smaller when compared to other two sources, however, this difference may be due to the natural variation in the climate and are not expected to change the main hydrological features in the lagoon.

[Figure]

**Figure S5.** Rose plot of wind condition in years of (a) 1970-1980, obtained from the Halls Head weather station; (b) 1981-2000, obtained from the WRF weather model; and (c) 2001-2016, obtained from the Mandurah weather station.

4. There is no calibration of the model parameters.

**Response:** We have now added a brief summary of the calibration approach in the revised manuscript (in section 2.2.3 - Model calibration, performance evaluation and sensitivity tests), and a supplemental document to this manuscript for the detailed results from the model calibration and validation, and the model sensitivity to selected environmental drivers.

5. Topography is also an important factor affecting the hydrology of an estuary. Between 1970 and 2016, how did the topography of the lagoon and the natural channel change and affect the hydrology? There is no anything about this.

**Response:** We agree that changes to the lagoon bathymetry could be a factor that could lead to changes in the hydrology of the estuary, and further explanation is required to clarify the potential significance of this. We had provided a literature review and discussion on this issue in our early response in the discussion forum. We have now included a new section in the discussion (section 4.3 - "Potential impacts from the morphological change and catchment development on the Peel-Harvey hydrology") in the revised manuscript to address the potential impacts from the morphological changes on the results.

6. The results for 2040 and 2060 based on projected climate seem to be too simple. And there is no any explanation.

**Response:** We acknowledge that our future climate projection is relatively simple to investigate the future hydrology in the Peel-Harvey Estuary, although our projections for weather and flow change were based on the average trend reported from more detailed studies using an ensemble of climate models (Silberstein et al., 2012; Smith and Power, 2014). The Peel-Harvey region has experienced a widely reported decline in rainfall over the last several decades (CSIRO & BoM 2007; IPCC 2007; CSIRO 2009; Hope & Ganter 2010). The trend in rainfall decline is expected to continue, based on the climate projections from general circulation models (GCMs) results (CSIRO 2009; Smith and Power, 2014). Given the nature of our research questions was to extrapolate the mean trend that we reported from the hind-cast simulations, we focus the future scenarios on the changes of hydrology under the projected average reduction in the flow from the ensemble models (Smith and Power, 2014), with an assumed mean rate of sea level rise (Kuhn et al., 2011), to highlight the general trend and allow for prioritization of adaptation strategies such as environmental water allocation policies. This approach is over-simplistic also in that it assumes no seasonal change in hydrologic trends, and there has been recent evidence that increasing summer floods are occurring and the winter peak flows are decreasing as a fraction of the annual total (McFarlane et al., 2020). We have now integrated this response into the manuscript to further explain the approach of future projection, plus added to the discussion on the significance of this uncertainty and the requirement of future research on this topic in section 4.4 – "Uncertainty of future hydrology".

7. In 4.1 section, the contents in paragraphs 1, 2 and 3 mostly repeated the results. Here, the compare between your results and other related research should be shown.

**Response:** Thank you for the suggestion. We focused our discussion on how the interaction of the climate change affects with the artificial channel on the hydrology in these paragraphs. We first compared results from the current study because study cases of the interaction of the climate and artificial channel are relatively rare. We have carried a further modelling work to study the mixing efficiency in the Peel-Harvey Estuary using the same numerical methods of water renewal time and flushing time as the studies of Umgiesser et al. (2014), who compared the impact of climate change on the hydrology of 10 Mediterranean lagoons, and improved the discussion of the water transportation and mixing in response to the changes in ocean connectivity and catchment flows by comparing our results to the findings from Umgiesser et al., (2014). We also improved the discussions of the individual and combined impacts of the drying climate and artificial channel on the lagoon hydrology evolution, and added discussions on the impacts from topographic changes and catchment development, and further explain the future climate projections.

**Minor points:**

1. All important stations or locations mentioned in the manuscript should be occurred in Figure 1.

**Response:** Comment accepted. We added the ungauged inflow locations, the weather station locations, and the site of Mandurah channel to the site map.

2. GL and mAHD should be changed to international units.

**Response:** Comment accepted. We have checked through the manuscript to use the SI units consistently. The unit of mAHD stands for elevation in meters with respect to the Australian Height Datum and is now explained in the Figure 1 caption.

3. Impacts on the stratification were shown in the results and conclusions. Why were they not mentioned in abstract?

**Response:** Thank you for pointing out this issue. We added the impacts on the stratification to the abstract as "The opening of the artificial channel is shown to increase the seawater fluxes and the salinity stratification, while the drying climate had reduced the salinity stratification in the main body of the estuary."

4. The citation of reference is disorder. If several references are cited together, they should be put in order according to publishing year.

**Response:** Thank you for pointing out this and apologize for the disorder of the reference citation. The citation was created automatically by a citation software in order of names. We have went through the manuscript and cleaned up the citation format.

5. In page 6, line 186, WA region means Western Australia? The indication of this abbreviation is not seen

**Response:** Yes the WA is an abbreviation for Western Australia. Sorry for missing the explanation of the abbreviation which is now added into the manuscript in the revision.

6. In Figure 1 "Peel Estuary" is indicated. However, in the text and other figures only "Peel Inlet" can be seen. Are they the same location? If yes, in Figure 1, text, and other figures they should be the same.

**Response:** Thank you for pointing out this. The Peel Inlet is also referred as Peel Estuary by local management agencies, but the formal name for this lagoon should be Peel Inlet. We apologize for the misuse of this name in the site map. We now use the name "Peel Inlet" consistently in the revision.

7. Page 10, lines 277-279, authors said "the tide elevations in the ocean showed similar characteristics in 1990 and 1998 in terms of the annual mean sea level (-0.071 mAHD and -0.027 mAHD in 1990 and 1998, respectively) and tidal range (both < 1 m)". The plot (a) of Figure 4 shows the detailed sea levels in 1990 and 1998. Why did you only compare the annual mean sea level? It can be seen from plot (a) that the sea levels in 1998 also had a wider range variation, similar to the estuary surface elevation.

**Response:** Thank you for pointing out this. We agree that the tide condition in the year of 1990 and 1998 need to be further declared. We have further analysed the exceeding percentage distribution of the tide elevations and compared the tide constituents in these two years, and added the results in section 3.1 – "Estuary response to the change in ocean connectivity".

8. Caption of Figure 4 is not proper. It looks like three figures.

**Response:** Comment accepted. We now combined the tidal elevation plot with the elevation exceedance plot in Figure 4, and showed the changes in surface elevation within the lagoon in Figure 6.

9. Table 3: (1) In caption of the table, "Summary" should be deleted. (2) About the performance of salinity, it can be seen that errors after 1998 are clearly larger. Why?

**Response:** (1) Comment accepted. We have now moved the model validation table to the supplementary document, and deleted the "Summary" from the caption. (2) We wonder the introduction of the artificial channel after 1994 may have increased the complexity of salinity in the 6 monitoring sites within the estuary, therefore introduce more bias in the model output when compared to observation. For example, mechanical sand bypassing has been undertaken in the Dawesville Cut each year to maintain the channel since construction. This operation may have affected the water exchange, however, cannot be resolved by the hydrological model.

10. Caption of Figure 10, "The darker symbols indicate the years with accidental summer rainfall events and caused the catchment inflows higher than 15 GL". In this sentence "and caused" seems syntax error.

**Response:** Yes it is a type error and thank you for pointing out this. We now reword the caption to be: "The darker symbols indicate the years with accidental summer rainfall events, during which the total catchment inflows in summer season higher than $15 \times 10^6$ m$^3$."